# Balancing Large-Scale Wildlife Protection and Forest Management Goals with a Game-Theoretic Approach

**Denys Yemshanov** [1,*], **Robert G. Haight** [2], **Ning Liu** [1], **Robert S. Rempel** [3], **Frank H. Koch** [4] **and Art Rodgers** [5]

1   Natural Resources Canada, Canadian Forest Service, Great Lakes Forestry Centre, 1219 Queen Street East, Sault Ste. Marie, ON P6A 2E5, Canada; ning.liu@canada.ca
2   USDA Forest Service, Northern Research Station, 1992 Folwell Ave., St. Paul, MN 55108, USA; robert.haight@usda.gov
3   FERIT Environmental Consulting, Thunder Bay, ON P7A 1P4, Canada; northernbio@gmail.com
4   USDA Forest Service, Southern Research Station, Eastern Forest Environmental Threat Assessment Center, P.O. Box 12254, Research Triangle Park, NC 27709, USA; frank.h.koch@usda.gov
5   Ontario Ministry of Natural Resources and Forestry, Centre for Northern Forest Ecosystem Research, 421 James St S., Thunder Bay, ON P7E 2V6, Canada; art.rodgers@ontario.ca
*   Correspondence: denys.yemshanov@canada.ca

**Abstract:** When adopted, wildlife protection policies in Canadian forests typically cover large areas and affect multiple economic agents working in these landscapes. Such measures are likely to increase the costs of timber for forestry companies operating in the area, which may hinder their acceptance of the policies unless harvesting remains profitable. We propose a bi-level wildlife protection problem that accounts for the profit-maximizing behavior of forestry companies operating in an area subject to protection. We consider the regulator with a wildlife protection mandate and forestry companies licensed to harvest public forest lands. We depict the relationship between the regulator and forestry companies as a leader-follower Stackelberg game. The leader sets the protected area target for each license area and the followers adjust their strategies to maximize payoffs while meeting the protection target set by the leader. The leader's objective is to maximize the area-wide protection of spatially contiguous habitat while accounting for the followers' profit-maximizing behavior. We apply the approach to investigate habitat protection policies for woodland caribou in the Churchill range, Ontario, Canada. We compare the game-theoretic solutions with solutions that do not consider the forest companies' objectives and also with solutions equalizing the revenue losses among the companies.

**Keywords:** Stackelberg game; bi-level optimization; network flow model; habitat connectivity; harvest scheduling model I; woodland caribou; equal revenue loss

## 1. Introduction

In boreal Canada (>50° N latitude), woodland caribou populations (*Rangifer tarandus caribou*) have been declining in areas of active forestry and resource extraction [1,2]. Woodland caribou is a threatened species in Canada [3] and requires protection across large forested landscapes that, historically, have been fragmented by these economic activities [4–6]. The most appropriate scale for the protection of woodland caribou populations is at the range level, where a range is an area of sufficient spatial extent to support a viable caribou population [7,8]; caribou ranges are commonly thousands of square kilometers in size [9]. Habitat protection within a range requires the creation of large undisturbed spaces that allow caribou to travel among critical areas, such as calving grounds [10–12], and are therefore essential for caribou persistence. Indeed, caribou constantly move over long distances as part of their foraging and seasonal behavior [13]. Therefore, all protected locations within the range must be connected by undisturbed corridors to limit exposure to predators. In general terms, whole-range protection can be envisioned as protecting an

interconnected pattern of forest habitats with the total area equal to a desired conservation management threshold (e.g., 65% of the range area).

In regions with active forest management, whole-range protection reduces the area of productive forest available for harvesting. This issue is an example of a "conservation conflict" (*sensu* [14]) when land use policies are subject to the pressures of competing objectives, such as prioritizing wildlife conservation and prioritizing timber production. The problem of balancing forestry operations and wildlife protection has been the focus of active research [15–24]. The typical approach is to combine a harvest planning problem and a set of wildlife habitat protection constraints in one linear programming problem that maximizes protection of the most suitable habitat while also allowing for the desired level of harvesting [20,23,24]. Combining the problems of harvest planning and habitat protection requires optimizing across forested landscapes with thousands of harvest blocks over multiple time periods and is computationally difficult [20,23,25]. Martin et al. [15] and Ruppert et al. [19] used a simpler approach by sequentially solving a harvest planning problem after estimating the patterns of suitable habitat with a heuristic model for woodland caribou. St. John et al. [20,21] combined a harvest scheduling model (following the Model I formulation; see [26]) with optimal selection of a wildlife corridor containing suitable reindeer habitat. Yemshanov et al. [23,24] linked a harvest scheduling model with a network flow problem to maximize the amount of connected habitat in a forested area while meeting a desired harvest volume target.

Most models that combine timber harvest and wildlife habitat protection objectives consider only one decision-making agent (e.g., one forestry company or one government agency) operating in a single forest management unit (e.g., [15,20]). In reality, multiple agents may operate in a wildlife range. For example, in Canada, a provincial government is mandated to protect wildlife habitat on public land and also issue licenses to multiple timber companies for timber harvesting on the same public land. In theory, the determination of protected habitat at the whole-range level can be guided by a "first-best" model, which prioritizes sites for protection starting from the most suitable habitat towards less suitable habitat until a desired conservation area threshold is met. However, this model ignores the differential impacts these habitat allocations can have on forestry companies with licenses to operate within the range. Assigning protected area targets to the portions of the range licensed to individual companies is challenging because the planners developing the whole-range conservation plans are unlikely to have complete knowledge of how habitat protection affects the profit margins of the individual companies. Furthermore, it may be difficult to secure cooperation among all companies to accept the habitat protection measures prescribed by the "first-best" model because the model ignores their individual interests, which will probably cause some companies to perceive the measures as unfair. As a result, forestry operations in Ontario were exempt from the province's Endangered Species Act for more than 5 years [27]. The problem of quantifying the responses of industry players to environmental regulation is not unique to forestry and has been acknowledged in other sectors, such as manufacturing [28,29] and energy production [30]. Ideally, an alternative approach to the "first-best" model would capture the profit-maximizing responses of the individual companies. In this study, we extend the problem of wildlife habitat protection in a region of active forestry by accounting for the profit-maximizing behavior of multiple forestry companies operating in the region.

We consider a regulatory problem in which a government agency with a mandate to protect an at-risk wildlife species (woodland caribou) must make spatial decisions to protect its habitat at the species range level. We recognize two categories of decision-makers: the regulator, which is a government agency with a mandate to protect the wildlife species' populations on behalf of the public, and forestry companies with licenses to harvest timber on public forest lands within predefined regions (forest management units, FMUs). In Canada, provincial government authorities issue these licenses, delimiting the forest management units where companies are granted harvest rights. The government also has statutory responsibility for caribou conservation, as an endangered species, according to the

Endangered Species Act [31,32]. The government agency can regulate harvest and habitat protection within the FMUs to protect habitat. We consider a decision-making problem of designing the regulator's habitat protection strategy so that it will be perceived as fair and equitable among the forestry companies. We depict such a regulatory strategy using a non-cooperative leader-follower Stackelberg game [33]. A Stackelberg model [34,35] involves a set of players who move sequentially. The leader moves first, and the other players (the followers) move second after observing the leader's move. Recently, the Stackelberg game concept was applied to investigate the impact of greenhouse gas emissions taxation on pricing decisions in remanufacturing [28]. It has also been used to extend a timber supply model to anticipate industrial fiber consumption from forest product manufacturers [36,37] and for allocating funds for fire suppression [38]. In our case, the leader (the regulator) sets the protected area proportion targets for each FMU in which a forest company (the follower) has a license to harvest timber. The followers select habitat areas to protect and adjust their harvesting plans in their licensed FMUs over a planning horizon of $T$ periods to maximize their profits while meeting the protected area proportion target set by the leader (the regulator).

A common approach to finding the optimal strategies in a Stackelberg game is to convert the game to an extensive-form representation of the leadership problem and discretize the space of pure strategies to which the follower can commit [39]. For every pure strategy of the follower, one needs to find the highest utility that the leader can obtain, under the condition that the leader plays a strategy in which the follower's strategy is the best response to the leader's move. In our model, the leader's objective is to maximize the amount of caribou habitat that is protected in the range. The optimal solution for the leader is to allocate the protected area proportion target for each license area (FMU) with anticipation of the followers' profit maximization responses to the protection measures set by the leader. Each follower maximizes their payoff by responding with a pure profit-maximizing strategy to the leader's commitment. If there are multiple optimal strategies for a follower, the main-level problem selects the one that maximizes the objective of the leader.

We solve this leader-follower problem using a bi-level formulation. Bi-level optimization is a common approach to solve hierarchical resource allocation problems [40], such as finding optimal government policies for biofuel production [41], transportation [42], biotechnology [43,44], chemical engineering [45,46], energy generation [47–49] and forestry [36,37,50–53]. We further constrain the bi-level problem to equalize the potential proportional losses among the forestry companies because of habitat protection. We illustrate the problem by analyzing the optimal protection strategies for the C\hurchill caribou range in northwestern Ontario, Canada (Figure 1), where we incorporate Ontario's current practices designed to reduce the impact of harvesting on caribou populations.

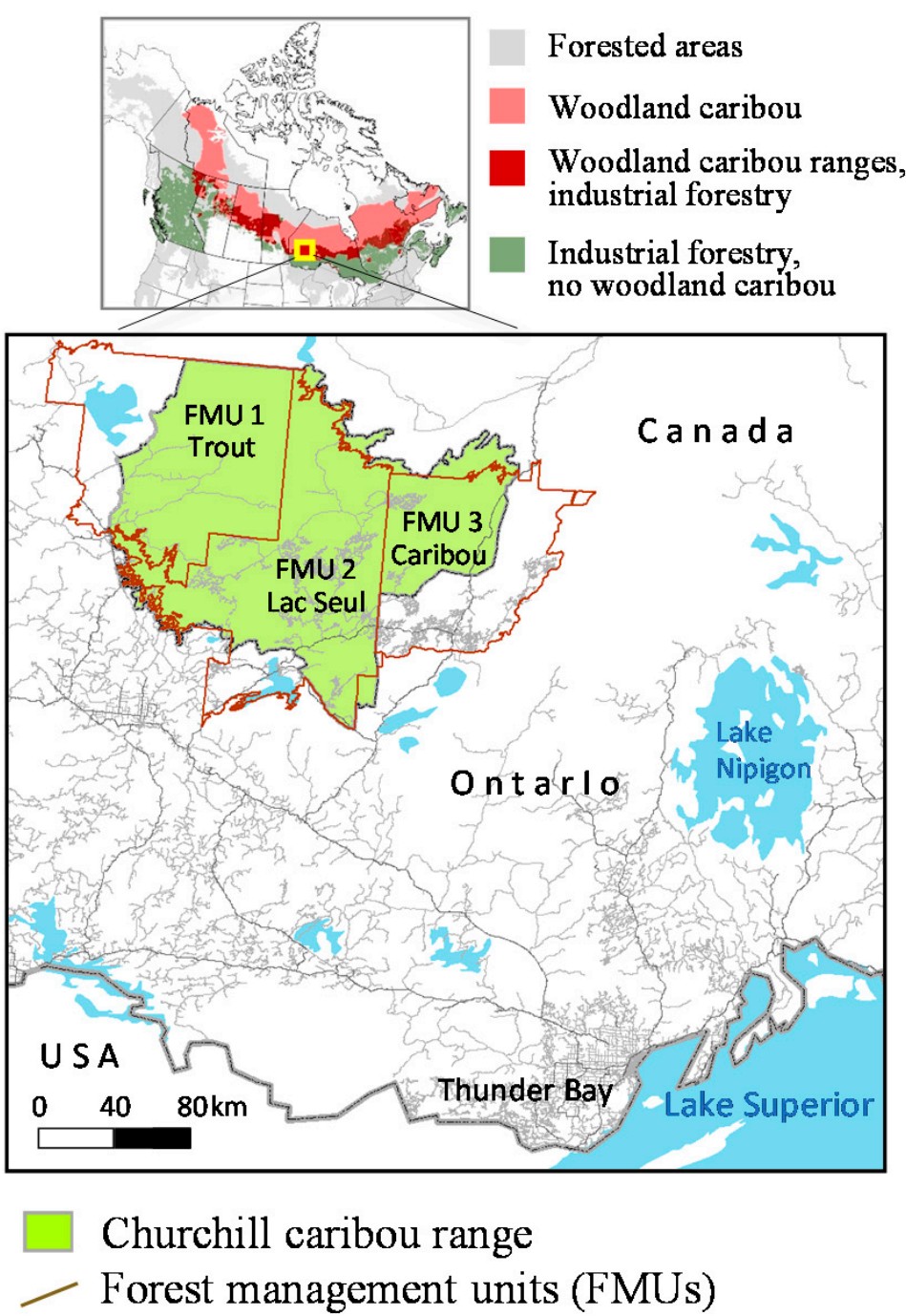

**Figure 1.** Case study area: Churchill caribou range, Ontario, Canada.

## 2. Materials and Methods

### 2.1. A Socially Optimal Habitat Protection Problem

We first formulate a whole-range habitat protection problem that does not consider the responses of forestry companies and maximizes the amount of protected habitat within the range, which we call the social optimum. Caribou require interconnected tracts of undisturbed habitat to shelter themselves from predators, therefore we need to ensure that all protected sites are connected. We formulate this patch connectivity problem using the network flow model from [24]. For a given upper bound on a protected habitat area, the network flow model chooses the patches to protect to maximize the amount of fully connected habitat. We depict the caribou range as a network of $J$ forest patches (nodes) that can support caribou. Each patch $j$ has an amount of caribou habitat $B_j$, which is

estimated as the product of habitat quality and habitat area in $j$. Individual patches must be sufficiently large to facilitate the movement of animals, which requires selecting the patch size compatible with the average daily distances covered by the animals [54–57]. We used a hexagonal grid of 6125-ha patches.

We conceptualize the protected area as a subnetwork of interconnected patches and assume that caribou can move between neighboring patches $k$ and $j$ if no human disturbances occur in $k$ and $j$ and both patches have suitable habitats. Our approach of controlling habitat connectivity with a network flow model follows from Sessions [58], who first proposed the formulation of the connected habitat problem as a Steiner network, and also relates to a network flow problem in [59]. We identify the potential movement of animals between patches $k$ and $j$ using a bi-directional pair of arcs $jk$ and $kj$. A binary variable $w_{kj}$ defines whether caribou individuals can move between patches $k$ and $j$ throughout the timespan $T$. A non-negative variable $y_{kj}$ characterizes the amount of flow through arc $kj$ connecting a pair of adjacent nodes $k$ and $j$. Controlling the amount of flow is required to control the connectivity of the habitat network.

We need to ensure that all patches selected for protection are connected and all unprotected patches are accessible for harvesting. This requires controlling the connectivity of subnetworks of both the protected and unprotected patches. Controlling connectivity between the unprotected patches prevents the creation of isolated pockets of unprotected forest surrounded by protected areas. Maintaining connectivity between undisturbed protected patches allows caribou to move and access their preferred habitat at a lower risk of encountering predators [60,61]. Connectivity between the protected (or unprotected) nodes $j$ can be achieved by injecting the flow into one protected node and ensuring that all other protected nodes receive flow from that node (or likewise for unprotected nodes) through the habitat network. Injection of the flow to a single node ensures that all nodes that receive flow from that node form a single connected graph.

We introduce an auxiliary node $j = 0$ to inject flow into the sub-networks of protected and unprotected patches to maintain their connectivity. Node 0 does not have a geographical location but is connected to all other nodes $j$. Node 0 is used to inject the flow into any selected node (Figure 2a) and ensures that all other selected nodes receive the flow (which ensures the connectivity of the subset of selected nodes). To avoid the creation of repetitive cycles between the connected patches, we assume that the flow can be fed to a connected node via, at most, one incoming arc.

A node (patch) $j$ is selected for protection if it receives incoming flow from any connected patch $k$ or node 0, i.e.,:

$$\sum_{k=0}^{\Theta_j} w_{kj} = 1 \tag{1}$$

where set $\Theta_j$ denotes all neighboring patches $k$, which are connected to patch $j$ and can transmit flow to $j$ (Figure 2b). A binary variable $v_{kj}$ defines the connection between the adjacent unprotected patches $k$ and $j$. A non-negative variable $z_{kj}$ characterizes the amount of flow through arc $kj$ connecting a pair of adjacent unprotected patches $k$ and $j$. The subnetworks of protected and unprotected patches use the same set of nodes $J$ but do not overlap (except node 0); a forest patch $j$ can be either a member of the subnetwork of protected or unprotected patches but not both. Table 1 lists the model parameters and variables.

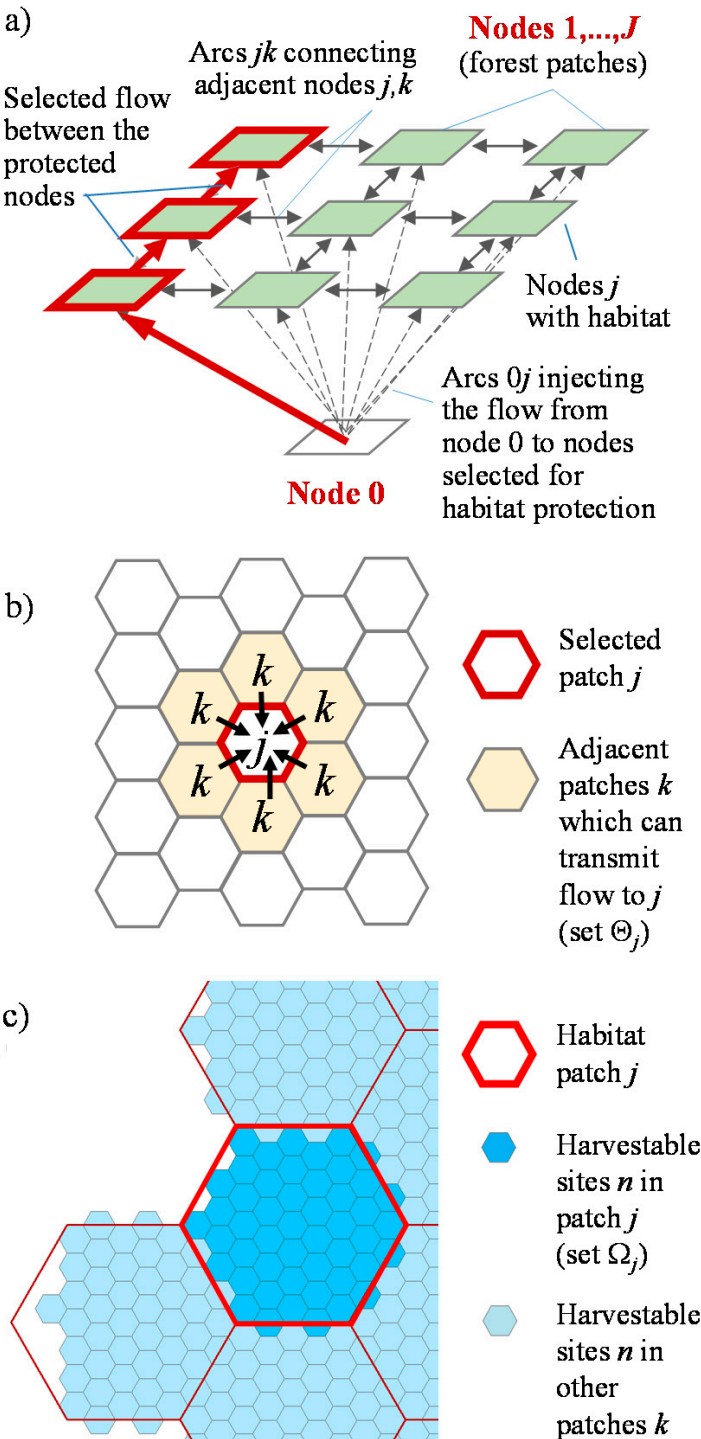

**Figure 2.** (**a**) The network flow model concept. Arrows show the universe of arcs, or connections, in the habitat network. Dashed arrows show connections from the root node 0 to nodes $j$ in the network. Root node 0 is used to inject the flow into the network. Arrows in bold red show examples of connected nodes with established flow from root node 0; (**b**) The concept of an adjacency set $\Theta_j$ around a selected node $j$; (**c**) Spatial alignment between the sets of harvestable sites $n$ and habitat patches $j$. Set $\Omega_j$ depicts harvestable sites $n$ located in habitat patch $j$. The selection of habitat patch $j$ for protection forbids harvesting all sites $n$ that belong to set $\Omega_j$.

**Table 1.** Summary of the model variables and parameters.

| Symbol | Parameter/Variable Name | Description |
|---|---|---|
| | Sets: | |
| $J$ | Patches $j$, $k$ with suitable habitat—potential candidates for protection in a landscape $J$ | $j,k \in J$ |
| $J_d$ | Patches $j$, $k$ with suitable habitat located in region $d$ | $J_d \in J$ |
| $\Theta_j$ | Patches $k$ which are connected to patch $j$ and can transmit flow to $j$ | $kj \in \Theta_j$ |
| $\Theta_j{}^+$ | Patches $k$ which are connected to patch $j$ and can receive flow from $j$ | $kj \in \Theta_j$ |
| $\Theta_{jd}$ | Patches $k$ located in region $d$ which can transmit flow to patch $j$ located in region $d$ | $\Theta_{jd} \in J$ |
| $N$ | Forest sites $n$—potential harvest blocks in landscape $N$ | $n \in N$ |
| $N_d$ | Forest sites $n$—potential harvest blocks located in region $d$ | $N_d \in N$ |
| $\Omega_j$ | Forest sites $n$—potential harvest blocks located in habitat patch $j$ | $\Omega_j \in N$ |
| $D$ | Forest management regions $d$ where individual players (the followers) operate | $d \in D$ |
| $C$ | Habitat protection levels $c$ | $c \in C$ |
| $T$ | Planning time periods, $t$ | $t \in T$ |
| $I$ | Harvest prescriptions, $i$ | $i \in I$ |
| | Decision variables: | |
| $w_{kj}$ | Binary indicator of the species flow via arc $kj$ in the protected area over timespan $T$ | $w_{jk} \in \{0,1\}$ |
| $v_{kj}$ | Binary indicator of the connection between the adjacent unprotected patches $k$ and $j$ | $v_{kj} \in \{0,1\}$ |
| $y_{kj}$ | Amount of flow between the adjacent protected patches $k$ and $j$ | $y_{jk} \geq 0$ |
| $z_{kj}$ | Amount of flow between the adjacent unprotected patches $k$ and $j$ | $z_{jk} \geq 0$ |
| $x_{ni}$ | Binary selection of harvest prescription $i$ in site $n$ | $x_{ni} \in \{0,1\}$ |
| $\omega_{dc}$ | Binary selection of the follower's optima from a solution set for $D$ regions $\times$ $C$ protection levels | $\omega_{dc} \in \{0,1\}$ |
| $\omega_{dc_{(S)}}$ | The value of binary variable $\omega_{dc}$ denoting the selection of the precomputed follower's solution $c$ with the protected area portion of FMU $d$ equal to selected target $S$. | $\omega_{dc_{(S)}} \in \{0,1\}$ |
| $P_1$ | The number of connections to node 0 above one in the network of protected patches | $P_1 \geq 0$ |
| $P_2$ | The number of connections to node 0 above $\theta$ in the network of unprotected patches | $P_2 \geq 0$ |
| $G$ | Proportional harvest revenue loss in region $d$ over timespan $T$ under protection level $c$ | $G \geq 0$ |
| $Q_d$ | Maximum sustainable harvest volume in region $d$ in period $t$ | $Q_d \geq 0$ |
| | Parameters | |
| $b_{nit}$ | Habitat amount in site $n$ in prescription $i$ in period $t$ | $b_{nit} \geq 0$ |
| $\beta_{nit}$ | Habitat suitability in site $n$ in prescription $i$ in period $t$ | $\beta_{nit} \geq 0$ |
| $B_j$ | Habitat amount in patch $j$ under protection (assuming the no-harvest prescription $i = 1$) | $B_j \geq 0$ |
| $\alpha_n$ | Habitat area in site $n$ | $\alpha_n \geq 0$ |
| $A_j$ | Habitat area in patch $j$ | $A_j \geq 0$ |
| $a_n$ | Harvestable forest area in site $n$ | $a_n \geq 0$ |
| $V_{nit}$ | Volume of timber available for the harvest at site $n$ in period $t$ in prescription $i$ | $V_{nit} \geq 0$ |
| $R_{ni}$ | Net cash flow associated with harvesting site $n$ according to prescription $i$ | $R_{ni} \geq 0$ |
| $E_{T\,\min}$ | Average target forest age in the managed area at the end of the planning horizon $T$ | 80 |
| $E_{ni}$ | Forest stand age in site $n$ at the end of the planning horizon if prescription $i$ is applied | 0–250 |
| $\rho_n$ | Net unit price of timber harvested from site $n$ | $\rho_n > 0$ |
| $e_n$ | Postharvest regeneration costs | $e_n > 0$ |
| $S_d$ | Target proportion of the protected area in region (FMU) $d$ | $S_d \in [0;1]$ |
| $S_c$ | Target proportion of the protected area at protection level $c$ | $S_c \in [0;1]$ |
| $S$ | Target proportion of the protected range area | $S \in [0;1]$ |
| $\chi_{jdc}$ | Binary selection of patch $j$ for protection in the follower's solution for region $d$ protection level $c$ | $\chi_{jdc} \in \{0,1\}$ |
| $r_{\max d}$ | Harvest revenue in the maximum sustainable harvest scenario without protection in region $d$ | $r_{\max d} \geq 0$ |
| $r_{dc}$ | Total harvest revenue over horizon $T$ in region $d$ at protection level $c$ | $r_{dc} \geq 0$ |
| $g_{dc}$ | Proportional loss in harvest revenue in region $d$ at protection level $c$ vs. no-protection scenario | $g_{dc} \in [0;1]$ |
| $\psi_j$ | Binary indicator of patches in northwestern and northeastern region's corners which must remain unharvested to maintain the connectivity of habitat between regions | $\psi_j \in \{0,1\}$ |
| $\gamma_{jd}$ | Binary parameter defining patches $j$ located in region $d$ | $\gamma_{jd} \in \{0,1\}$ |
| $\delta_{nd}$ | Binary parameter defining sites $n$ located in region $d$ | $\delta_{nd} \in \{0,1\}$ |
| $\mu_j$ | Binary parameter defining sites where permanent roads enter the area | $\mu_j \in \{0,1\}$ |
| $\theta$ | The number of major access points of entry to the range area (equal to the number of regions $D$) | 3 |
| $\sigma$ | Allowable decrease in harvest volume in consecutive planning periods $t$ and $t+1$ | 0.02 |
| $\varepsilon$ | Small value | 0.05 |
| $f_1, f_2, f_3$ | Scaling factors | $f_1, f_2, f_3 > 0$ |
| $U$ | Large positive value | $U > 0$ |

We formulate the habitat protection problem for the central regulator (problem 1 hereafter) by selecting a connected subnetwork of patches for habitat protection that

maximizes the amount of connected habitat in the protected patches weighted by its quality, subject to a target proportion of the area of patches to protect $S$, $S \in [0;1]$, i.e.,:

$$\max \sum_{j=1}^{J} \sum_{k=0}^{\Theta_j} (w_{kj} B_j) - f_1 P_1 - f_2 P_2 \tag{2}$$

s.t.:

$$\sum_{k=0}^{\Theta_j} y_{kj} - \sum_{k=1}^{\Theta_j^+} y_{jk} = \sum_{k=0}^{\Theta_j} w_{kj} \ \forall \ j \in J \tag{3}$$

$$\sum_{k=0}^{\Theta_j} z_{kj} - \sum_{k=1}^{\Theta_j^+} z_{jk} = \sum_{k=0}^{\Theta_j} v_{kj} \ \forall \ j \in J \tag{4}$$

$$y_{kj} \leq U w_{kj} \ \forall \ j \in J, k \in \Theta_j \tag{5}$$

$$w_{kj} \leq y_{kj} \ \forall \ j \in J, k \in \Theta_j \tag{6}$$

$$z_{kj} \leq U v_{kj} \ \forall \ j \in J, k \in \Theta_j \tag{7}$$

$$v_{kj} \leq z_{kj} \ \forall \ j \in J, k \in \Theta_j \tag{8}$$

$$P_1 \geq \sum_{j=1}^{J} w_{0j} - 1 \tag{9}$$

$$P_2 \geq \sum_{j=1}^{J} v_{0j} - \theta \tag{10}$$

$$\sum_{k=0}^{\Theta_j} v_{kj} + \sum_{k=0}^{\Theta_j} w_{kj} = 1 \ \forall \ j \in J \tag{11}$$

$$\sum_{k=0}^{\Theta_j} v_{kj} \geq \mu_j \ \forall \ j \in J | \mu_j > 0 \tag{12}$$

$$\sum_{k=0}^{\Theta_j} w_{kj} \leq 1 \ \forall \ j \in J \tag{13}$$

$$\sum_{k=0}^{\Theta_j} v_{kj} \leq 1 \ \forall \ j \in J \tag{14}$$

$$(S - \varepsilon) \sum_{j=1}^{J} A_j \leq \sum_{j=1}^{J} \sum_{k=0}^{\Theta_j} (w_{kj} A_j) \leq (S + \varepsilon) \sum_{j=1}^{J} A_j \tag{15}$$

The first term in objective (2) denotes the total aggregate amount of connected habitat across the range area. Penalty $P_1$ defines the number of connections to node 0 (and correspondingly the maximum number of separate contiguous subgraphs) above one in the subnetwork of protected patches. Penalty $P_2$ defines the number of connections to node 0 in the subnetwork of the unprotected patches above $\theta$, which is the number of major access points via roads entering the range. In our study, we assume one major access point to each FMU. Scaling weights $f_1$ and $f_2$ for the penalties $P_1$ and $P_2$ control their relative impacts on the objective value and are chosen so that the $P_1 f_1$ and $P_2 f_2$ terms are close to zero in the optimal solution. Penalty formulation works better than a hard constraint when landscape configuration does not allow for the creation of a single protected area, or when the FMU is accessible from multiple entry points.

Constraints (3) and (4) describe the flow balance through patch $j$ and ensure the connectivity of the subnetworks of protected (Equation (3)) and unprotected patches

(Equation (4)). The amount of flow coming to patch *j* is equal to the amount of outgoing flow from *j* plus the fulfilled capacity at *j* (one unit of flow). Set $\Theta_j$ denotes patches *k* which are connected to patch *j* and can transmit flow to *j* (Figure 2b), and set $\Theta_j^+$ denotes patches *k* which are connected to patch *j* and can receive flow from *j*. Constraints (5)–(8) ensure agreement between the amount of flow through arc *kj* in the subnetworks of protected and unprotected patches, and the arc selection variables $w_{kj}$ and $v_{kj}$. Constraints (5) and (7) ensure that the flow between patches *k* and *j* is zero when arc *kj* is not selected. Constraints (6) and (8) ensure that the arc selection variables $w_{kj}$ and $v_{kj}$ are zero unless a positive flow is established from node *k* to node *j*. Constraints (9) and (10) define penalties $P_1$ and $P_2$. Constraint (11) specifies that patch *j* can only be a member of the subnetwork of protected or unprotected sites but not both. Constraint (12) ensures that the unprotected area includes patches where major roads enter the area. A binary parameter $\mu_j$ defines the sites where permanent roads enter the area ($\mu_j = 1$ and $\mu_j = 0$ otherwise). Constraints (13) and (14) ensure that the flow in the subnetworks of protected and unprotected patches comes to a given patch *j* from at most one source. Constraint (15) sets the target proportion of the protected range area, $S \pm \varepsilon$, $S \in [0;1]$. Symbol $\varepsilon$ defines a small value, $\varepsilon = 0.05$ and $A_j$ defines the area of patch *j*.

*2.2. A Bi-Level Habitat Protection Problem*

In the bi-level habitat protection problem (problem 2, Table 2), we divide the landscape into *D* forest management units, (FMUs) where different forest companies (the followers) have licenses to harvest timber (Figure 1). For each FMU *d*, $d \in D$, the regulator sets the target proportion of the FMU's area that must be protected, $S_d$. In their respective FMUs *d*, the followers maximize their profits from harvesting timber under a set of harvest sustainability constraints and the obligation to protect the target area proportion $S_d$ set by the regulator. The regulator (the leader in the Stackelberg game) anticipates the profit-maximizing behavior of the followers when allocating the protected area proportion targets to each FMU.

**Table 2.** Habitat protection and harvest problems *.

| Problem Type and Objective | Equalization Constraints for FMUs *d* | | |
| --- | --- | --- | --- |
| | **None** | **Equal Protected Area Proportion** | **Equal Proportional Harvest Revenue Loss** |
| Bi-level, max (protected habitat) | Bi-level habitat protection problem 2 | Bi-level equal protected area proportion problem 4 | Bi-level equal proportional revenue loss problem 6 |
| Single-level, socially optimal max (protected habitat) | Socially optimal habitat protection problem 1 | Socially optimal habitat protection equal protected area proportion problem 3 | Socially optimal habitat protection equal proportional revenue loss problem 5 |
| Single-level, max (harvest revenue) | Maximum harvest revenue problem 7 | | |

* All problems enforce the protection of a target range area proportion $S \pm \varepsilon$.

We solve the leader's problem using backward induction. We discretize all possible allocations of the protected area proportion in each FMU *d* to *C* candidate levels, $S_1, \ldots, S_C$. The first step solves each follower's problem for all possible protection levels *c*, $c \in C$, which can be set by the regulator for a given FMU. Then, based on the optimal solution for each FMU *d* and protection level *c*, we define a set of binary parameters $\chi_{jdc}$ for all patches *j* for whether or not patch *j* was selected for protection (i.e., $\chi_{jdc} = 1$ when patch *j* is selected for protection in the follower's solution in FMU *d* at protection level *c*, and $\chi_{jdc} = 0$

otherwise). In the second step, the regulator allocates the protected area proportion targets to the FMUs $d$ to maximize the leader's objective, assuming that each follower will use their profit-maximizing response strategy and the corresponding protected site pattern $\chi_{jdc}$ under the protection level $c$. This is a backward induction procedure because the followers' problems are solved first to create the response solutions and the regulator's problem is solved second given the response solutions of the followers.

*2.3. The Follower's Problem*

For each FMU $d$, we depict the follower's profit-maximizing strategy by solving a harvest planning problem for the subset of patches outside of the protected area within $d$. We apply harvest scheduling model 1 [26], which considers $N_d$ forest sites that could be harvested for timber in FMU $d$ over a planning horizon $T$. Harvest planning is done at the level of cut blocks that are smaller than the typical size of patches considered for wildlife protection. We depict these cut blocks using a finer-scale hexagonal grid $N$ of forest sites $n$ than the network of $j$ habitat patches. Each patch $j$ with caribou habitat includes a set $\Omega_j$ of smaller sites $n$, $n \in \Omega_j$, that represent potential cut blocks of a size consistent with large clear-cuts in northern boreal forests (Figure 2c). We only consider clear-cut harvest, which is the prevailing harvest type in boreal Canada [62]. If a patch $j$ is in the unprotected portion of FMU $d$, all forest sites $n$ in $j$, $n \in \Omega_j$ can be harvested after trees reach a minimum age (70 years in this study). For each site $n$, we define a set of prescriptions $i$, $i = 1, \ldots, I$, with possible sequences of harvest events over a horizon of $T$ planning periods in $n$, including a sequence $i = 1$ without harvest. A binary variable $x_{ni}$ selects whether site $n$ follows prescription $i$ with a defined sequence of harvest times (or no harvest). A site $n$ can be assigned one prescription only.

Clear-cut harvesting temporarily degrades the caribou habitat as it reduces the number of local foraging resources, and early successional vegetation attracts deer (*Odocoileus virginianus*) and moose (*Alces americanus*) which are followed by predators (black bears (*Ursus americanus*) and wolves (*Canis lupus*)) that also prey upon caribou [63–65]. The suitability of the caribou habitat improves 35–60 years after harvest as the forest stands mature [66–68]. Hence, the amount of habitat in a forest site depends on the time since the last harvest. For each site $n$ we define the amount of caribou habitat, $b_{nit}$ in period $t$ under prescription $i$ as a product of the habitat suitability $\beta_{nit}$ in period $t$ under prescription $i$ and the habitat area, $\alpha_n$, in site $n$: $b_{nit} = \alpha_n \beta_{nit}$. For period $t$, the amount of habitat in patch $j$ is the sum of the habitat amounts in all sites $n$ located in $j$, i.e.,:

$$\sum_{n=1}^{\Omega_j} \sum_{i=1}^{I} b_{nit} x_{ni} \tag{16}$$

Over horizon $T$, no harvesting is allowed in patches $j$ selected for protection. Recall that in the set of harvest prescriptions $I$, the scenario without harvest is enumerated as prescription $i = 1$. Thus, the amount of habitat in a protected patch $j$, $B_j$, is computed as the sum of the habitat amounts in all sites $n$ located in the patch under the no-harvest prescription $I = 1$ over $T$ periods, i.e.,:

$$B_j = \sum_{n=1}^{\Omega_j} \sum_{t=1}^{T} b_{n1t} \tag{17}$$

Each site $n$ includes one forest stand characterized by average age, forested area $a_n$, the volume of timber per unit area, $V_{nit}$, that could be harvested in period $t$ under prescription $i$ and the amount of habitat, $b_{nit}$ in period $t$ under prescription $i$.

The follower's problem finds the optimal habitat connectivity pattern for a given FMU. For each FMU $d$ and habitat protection level $c$, the follower's problem finds a connected subnetwork of protected patches under the target protected area proportion $S_d$

by maximizing the net cash flow, $R_{ni}$, from harvesting unprotected forest patches in $d$ over $T$ periods, i.e.,:

$$\max \sum_{n=1}^{N} \sum_{i=1}^{I} R_{ni} x_{ni} - f_1 P_1 - f_2 P_2 \tag{18}$$

s.t.: constraints (3)–(12) and

$$\sum_{k=1,\{0\}}^{\Theta_j} w_{kj} \leq \gamma_{jd} \ \forall j \in J \tag{19}$$

$$\sum_{k=1,\{0\}}^{\Theta_j} v_{kj} \leq \gamma_{jd} \ \forall j \in J \tag{20}$$

$$\sum_{k=0}^{\Theta_j} w_{kj} \geq \psi_j \ \forall j \in J | \psi_j > 0 \tag{21}$$

$$(S_c - \varepsilon) \sum_{j=1}^{J} A_j \leq \sum_{j=1}^{J} \sum_{k=0}^{\Theta_j} (w_{kj} A_j) \leq (S_c + \varepsilon) \sum_{j=1}^{J} A_j \tag{22}$$

$$\sum_{n=1}^{\Omega_j} \sum_{i=1}^{I} \sum_{t=1}^{T} (x_{ni} V_{nit})/U + \sum_{k=0}^{\Theta_j} w_{kj} \leq 1 \ \forall j \in 1, \ldots, J \tag{23}$$

$$\sum_{i=1}^{I} x_{ni} = 1 \ \forall n \in N_d \tag{24}$$

$$(1 - \sigma) Q_d \delta_{nd} \leq \sum_{n=1}^{N_d} \sum_{i=1}^{I} a_n V_{nit} x_{ni} \leq Q_d \delta_{nd} \ \forall t \in T \tag{25}$$

$$\sum_{n=1}^{N_d} \delta_{nd} \left( \sum_{i=1}^{I} [(E_{ni} - E_{T\min}) a_n x_{ni}] \right) \geq 0 \tag{26}$$

The follower's model is solved for all combinations of $D$ FMUs $\times$ $C$ protection levels. For consistency, we formulate the problem for the whole range area but use masking constraints to restrict the problem to an individual FMU. Given that the habitat protection problem does not use discounting, we used the undiscounted harvest revenue flows to ensure similar handling of the habitat protection and harvesting problems. The harvest revenue $R_{ni}$ in Equation (18) is calculated as the value of delivered timber (at the mill gate) net of harvest, hauling, and post-harvest regeneration costs, $\sum_{t=1}^{T}(a_n \rho_n V_{nit} - e_n)$ where $\rho_n$ is the net unit price of timber harvested from site $n$ and $e_n$ is the post-harvest regeneration cost in $n$.

Constraints (3)–(12) are the same as in problem 1 except that the $\theta$ value in constraint (10) is set to 1 (i.e., assuming at least one road entry per FMU). Constraints (19) and (20) restrict the habitat connectivity problem to an FMU $d$ and ensure that the flow to patch $j$ in the subnetworks of protected and unprotected patches can come from one source and only patches located in $d$ are considered, as indicated by a binary parameter $\gamma_{jd}$ ($\gamma_{jd} = 1$ if a patch $j$ is in FMU $d$, and $\gamma_{jd} = 0$ otherwise). Constraint (21) ensures that the protected patches in FMU $d$ are connected to at least one of a small number of patches located in the northwestern and/or northeastern corner of each FMU (yellow patches in Figure 3a). A binary parameter $\psi_j$ defines these patches, which are expected to stay unharvested due to poor access ($\psi_j = 1$ and $\psi_j = 0$ otherwise). Including these patches in the protected subnetwork guarantees the connectivity of the protected habitat patterns between adjacent FMUs. Constraint (22) sets the target proportion of the protected range area at level $c$, $S_c \pm \varepsilon$, and constraint (23) specifies no harvesting in protected patches. Together, constraints (3)–(12) and (19)–(21) ensure the spatial connectivity of the subnetworks of protected and unprotected patches both within and between the FMUs.

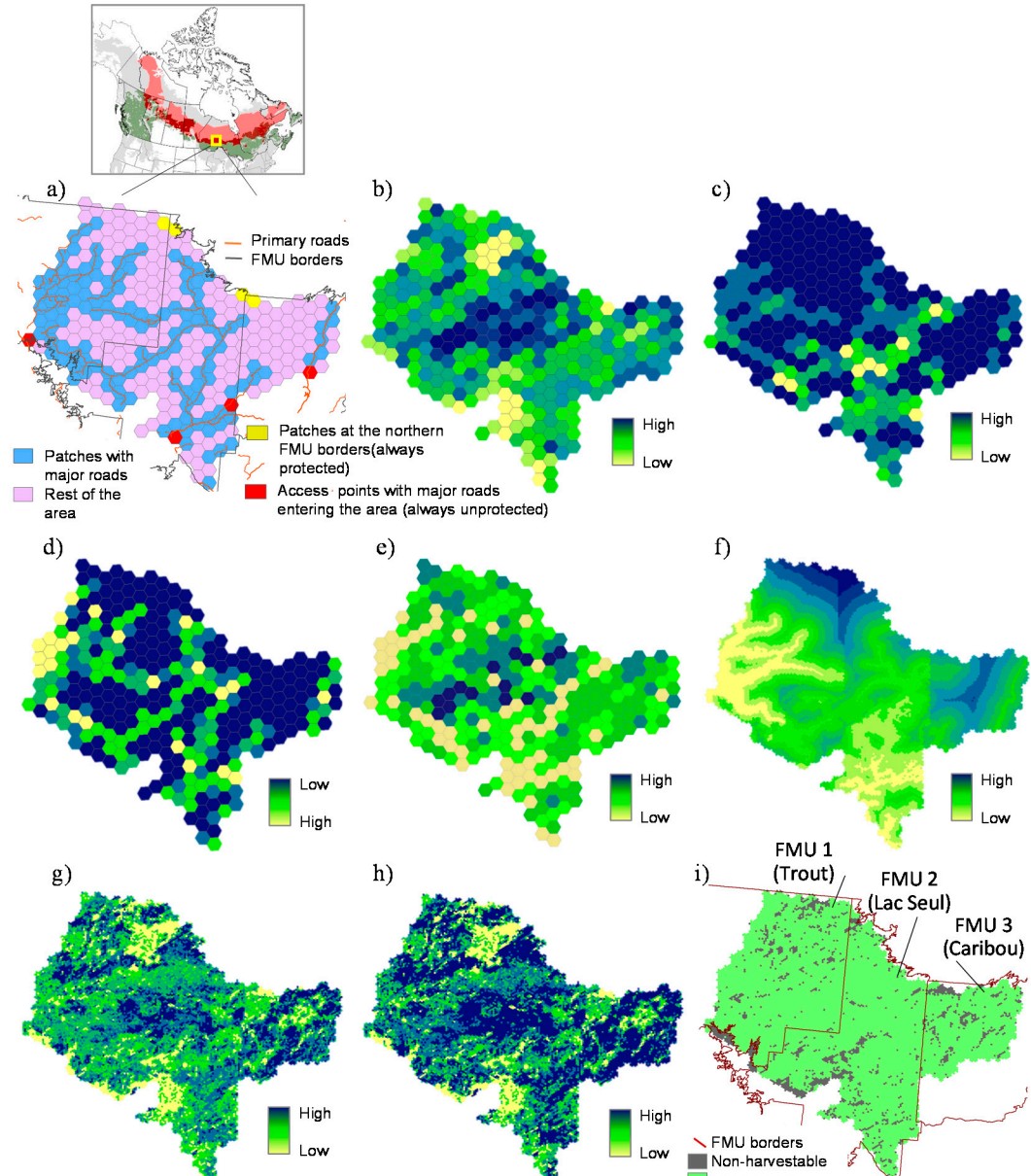

**Figure 3.** Model spatial inputs: (**a**) Patches *j* with permanent roads, access points with major roads entering the area and patches near the northern range border and the corners of adjacent regions that must remain protected to maintain habitat connectivity across the range; (**b**) habitat suitability map, $\beta_{nit}$ at *t* = 0; (**c**) intactness values; (**d**) permanent roads impact factor; (**e**) habitat amounts $B_j$; (**f**) timber hauling cost; (**g**) forest age at *t* = 0; (**h**) harvestable timber volume at *t* = 0; (**i**) harvestable area within FMUs 1–3.

As noted previously, the harvest scheduling and habitat connectivity models operate at different spatial resolutions (Figure 2c). The habitat connectivity constraints (3)–(12) and (19)–(22) are applied to a set of patches *J* but the harvest scheduling constraints (24)–(26) are applied to a set of harvestable sites *N*. Constraint (23) specifies agreement between the selection of harvest prescriptions for sites *n* in set *N* and the selection of patches *j* for protection in set *J*. The first term in (23) is the total volume harvested in sites *n* located in patch *j*, *n* ∈ $\Omega_j$, divided by a large value, *U*, to keep it within a [0;1] range. The second term defines the selection of patch *j* for protection. Constraint (24) ensures that each site *n* is assigned one prescription only. For each FMU *d*, we define a non-negative variable, $Q_d$, that sets the maximum sustainable harvest volume in period *t*. Constraint (25) ensures that the harvested volume for each period *t* stays close to a maximum sustainable range within

$[Q_d (1 - \sigma); Q_d]$, where $\sigma$ is a small value, $\sigma = 0.02$. The binary parameters $\delta_{nd}$ in constraints (25) and (26) defines sites $n$ located in FMU $d$ ($\delta_{nd} = 1$ and $\delta_{nd} = 0$ otherwise). Constraint (25) enforces even harvest flow over horizon $T$ and no harvest outside of $d$. Constraint (26) sets the average age of forest stands at the end of $T$ in FMU $d$ to be greater than or equal to the minimum age target $E_{T \min}$, which prevents overharvesting. Parameter $E_{ni}$ defines the forest stand age in site $n$ at the end of horizon $T$ if prescription $i$ is applied.

After solving the follower's problem for all combinations of $D$ FMUs $\times$ $C$ protection levels, we store the locations of patches $j$ selected for protection using a binary parameter $\chi_{jdc}$, i.e.,:

$$\chi_{jdc} = \sum_{k=0}^{\Theta_j} w_{kj} \ \forall \ j \in J, d \in D, c \in C \tag{27}$$

For each patch $j$ in FMU $d$ at protection level $c$, Equation (27) calculates the sum of arcs with the incoming flow to $j$. Because constraint (19) limits the flow to patch $j$ to no more than one arc in FMU $d$, the parameter $\chi_{jdc}$ can be either 1, when the node $j$ is protected, or 0 otherwise.

### 2.4. The Leader's Habitat Protection Problem

The leader's part of the bi-level habitat protection problem (problem 2 hereafter) finds an optimal pattern of protected habitat by selecting, in each FMU $d$, between the discretized habitat protection levels $c$, whose corresponding profit-maximizing optima have been precomputed by solving the follower's problem for all $D \times C$ combinations. For each patch $j$, a binary variable $\omega_{dc}$ selects the precomputed optimum (and a corresponding subset of the protected patch locations $\chi_{jdc}$) from a solution set for $D$ FMUs and $C$ protection levels. The objective function (28) is similar to the problem 1 objective and maximizes the total amount of connected habitat in the protected area under the protected area proportion target $S$, i.e.,:

$$\max \sum_{j=1}^{J} \sum_{k=0}^{\Theta_j} (w_{kj} B_j) - f_1 P_1 - f_2 P_2 \tag{28}$$

s.t.: constraints (3)–(15) and:

$$\sum_{k=1,\{0\}}^{\Theta_j} w_{kj} = \sum_{d=1}^{D} \sum_{c=1}^{C} \omega_{dc} \chi_{jdc} \ \forall \ j \in 1, \dots, J \tag{29}$$

$$\sum_{c=1}^{C} \omega_{dc} = 1 \ \forall \ d \in D \tag{30}$$

where

$$\chi_{jdc} \in \underset{D,C}{\operatorname{argmax}} \left( \sum_{n=1}^{N} \sum_{i=1}^{I} R_{ni} x_{ni} - f_1 P_1 - f_2 P_2, \text{ s.t. : } \text{constraints } (3) - (12), (19) - (26) \right) \tag{31}$$

Constraints (3)–(15) ensure the spatial connectivity of the subnetworks of protected and unprotected patches and are the same as in the socially optimal habitat protection problem 1. Constraint (29) restricts the selection of the protected patches to a configuration prescribed by one of the precomputed follower's optima. Constraint (30) ensures the selection of the precomputed follower's solution in FMU $d$ for one protection level $c$ only. Constraint (15) enforces the selection of precomputed optima such as the total proportion of the protected range area meets the protection target $S \pm \varepsilon$. Equation (31) defines a binary parameter $\chi_{jdc}$ with the locations of protected patches as based on a set of precomputed optima in the follower's problem.

### 2.5. Equal Protected Area Proportion Problem

Problems 1 and 2 do not control the amount of protected area in a given FMU within the range. Some forestry companies may perceive unequal FMU allocations of the protected area as unfair. We evaluated problems 1 and 2 with added constraints to protect equal area proportions in each FMU. The equal protected area proportion problem 3 follows the problem 1 formulation except for the protection target constraint (15) is replaced with constraint (32), which sets separate protected area proportion targets, $S \pm \varepsilon$, for each FMU $d$:

$$(S - \varepsilon) \sum_{j=1}^{J_d} A_j \leq \sum_{j=1}^{J_d} \sum_{k=0}^{\Theta_{jd}} (w_{kj} A_j) \leq (S + \varepsilon) \sum_{j=1}^{J_d} A_j \; \forall \, d \in D \tag{32}$$

Set $J_d$ denotes patches $j$ located in FMU $d$ and set $\Theta_{jd}$ denotes patches in FMU $d$ which can transmit flow to patch $j$ located in FMU $d$. The bi-level equal protected area proportion problem 4 is based on the problem 2 formulation with the protected area proportion constraint (15) replaced by constraint (33), which selects, for each FMU $d$, the precomputed follower's solution at the protection level equal to the target area proportion $S$, i.e.,:

$$\omega_{dc_{(S)}} = 1 \; \forall \, d \in D \tag{33}$$

The value of the binary variable $\omega_{dc_{(S)}}$ defines the precomputed follower's solution $c$ with the protected portion of the FMU area $d$ equal to a target value $S$.

### 2.6. Equal Proportional Revenue Loss Problem

We also evaluated alternative formulations where habitat protection is allocated to equalize the proportional revenue losses among followers after the protection policy is enacted. We first solved, for each FMU $d$, the harvest planning problem with no protection, when the harvest is only limited by environmental sustainability constraints (25) and (26). For each FMU, this harvest solution yields the revenue value without protection, $r_{\max d}$. Then, we modified the socially optimal and bi-level habitat protection problems 1 and 2 to enforce equal harvest revenue losses in each FMU. The equal proportional revenue loss problem 5 maximizes the weighted sum of the protected habitat amount and the rescaled harvest revenue, i.e.,:

$$\max \sum_{j=1}^{J} \sum_{k=0}^{\Theta_j} (w_{kj} B_j) - f_1 P_1 - f_2 P_2 + f_3 \sum_{n=1}^{N} \sum_{i=1}^{I} R_{ni} x_{ni} \tag{34}$$

s.t.: constraints (3)–(15), (23) and

$$\sum_{i=1}^{I} x_{ni} = 1 \; \forall \, n \in N \tag{35}$$

$$(1 - \sigma) Q_d \delta_{nd} \leq \sum_{n=1}^{N_d} \sum_{i=1}^{I} a_n V_{nit} x_{ni} \leq Q_d \delta_{nd} \; \forall \, t \in T, d \in D \tag{36}$$

$$\sum_{n=1}^{N_d} \delta_{nd} \left( \sum_{i=1}^{I} [(E_{ni} - E_{T\min}) a_n x_{ni}] \right) \geq 0 \; \forall \, d \in D \tag{37}$$

$$G - \varepsilon \leq \frac{\left( r_{\max d} - \sum\limits_{n=1}^{N} \sum\limits_{i=1}^{I} R_{ni} x_{ni} \right)}{r_{\max d}} \leq G + \varepsilon \; \forall \, d \in D \tag{38}$$

We set the scaling factor for harvest revenues, $f_3$, in the objective function (35) to a low value (0.001) to emphasize the protection of habitat. The harvest revenue term in objective (35) was required to ensure correct behavior of the harvest scheduling sub-problem

(Equations (23), (35)–(37)). Constraints (3)–(15) are based on the problem 1 formulation and constraints (35)–(37) and are analogous to the harvest scheduling constraints (24)–(26) in the follower's problem within problem 2, except that constraints (35)–(37) are applied to each FMU $d$. Constraint (38) ensures equal proportional revenue loss within $G \pm \varepsilon$ limits for all FMUs. A non-negative decision variable $G$ in (38) defines the proportional loss in harvest revenue in each FMU after imposing the protection at level $c$.

Problem 2 was modified as follows to incorporate equal revenue loss. For each precomputed follower's solution in FMU $d$ and protection level $c$, we calculated the proportional harvest revenue loss versus a no-protection harvest scenario, $g_{dc}$, as $(r_{\max d} - r_{dc})/r_{\max d}$, where symbol $r_{dc}$ defines the total harvest revenue in FMU $d$ at habitat protection level $c$. The bi-level equal proportional revenue loss problem 6 is based on the problem 2 formulation (Equations (3)–(15), (28)–(31)) with an extra constraint [39] that enforces the selection of the precomputed follower's optima with equal proportional revenue losses $G \pm \varepsilon$ for all FMUs, i.e.,:

$$G - \varepsilon \le \sum_{c=1}^{C} (\omega_{dc} g_{dc}) \le G + \varepsilon \ \forall \ d \in D \tag{39}$$

### 2.7. Maximum Harvest Revenue Problem

Additionally, we evaluated the range-wide maximum harvest revenue solution for a given habitat protection target $S$. The maximum harvest revenue problem 7 uses the habitat connectivity constraints (3)–(15) from problem 1 and the harvest planning constraints (35)–(37) from problem 5. The objective function (18) maximizes total harvest revenue, subject to constraints (3)–(15), (23), (35)–(37).

We explored the trade-off between maximizing the harvest revenue in problem 7 and maximizing the amount of protected habitat in problem 1. The objective function is a weighted average between the problem 1 and 7 objectives and uses the problem 5 objective function (34)), subject to constraints (3)–(15), (23), (35)–(37). We constructed the trade-off frontier by solving the problem for different values of the scaling factor $f_3$ between 0.001 and 10. Table 2 summarizes problems 1–7.

We composed the model in the General Algebraic Modeling System (GAMS v.33) [69] and solved it with the GUROBI linear programming solver [70]. In the bi-level problems 2, 4, and 6, the follower's problems reached optimality gap values below 0.01 in 3–6 h, except the solutions for FMUs 2 and 3 with $S$ targets 0.75–0.85, which required 11–24 h to attain gap values below 0.1. The subsequent leader's problems took less than 30 min to solve. The socially optimal habitat protection problems 1 and 3 took between 6 and 12 h to reach optimality gap values of 0.05 or less, while problem 5 took 34–48 h to reach gap values of 0.05 or less.

### 2.8. Case Study

We applied models 1–7 to assess whole-range habitat protection options in the Churchill caribou range in northwestern Ontario [71,72] (Figure 1). The area is adjacent to Wabakimi Provincial Park and includes three forest management units (Trout Lake Forest, Lac Seul Forest, and Caribou Forest) that all experience moderate levels of industrial harvesting. Each FMU is licensed to a major timber company: Trout Lake Forest to Domtar Pulp and Paper Products Inc. (Montreal, QC, Canada), Lac Seul Forest to Ondaadiziwin Forest Management Inc. (Hudson, ON, Canada), and Caribou Forest to Resolute Forest Products Inc. (Montreal, QC, Canada), with timber delivered to pulp and paper mills in the nearby cities of Thunder Bay and Dryden, ON.

### 2.9. Data

Each 6125-ha patch $j$ in our study area landscape was divided into a set of smaller 125-ha hexagons (Figure 2c) which represent harvestable sites $n$ located within $j$. For each site $n$, we estimated the habitat amounts $b_{nit}$ for each harvest prescription and period $t$ using the caribou habitat model for Ontario's Northwest Region [73], which considers

both land cover composition and forest stand age; based on these criteria, locations are designated as unsuitable, useable, preferred or refuge habitat (Table 3).

**Table 3.** Minimum age (years) at which a stand attains suitable caribou habitat status.

| Land Cover Type | Habitat Type | | |
|---|---|---|---|
| | **Useable** | **Preferred** | **Refuge** |
| Lowland spruce | | | 61 |
| Mixedwood conifers | | | 71 |
| Other lowland conifers | 51 | | 41 * |
| Jack pine dominant | 41 | 61 | 41 * |
| Jack pine mixed wood | 41 | 61 | 41 |
| Black spruce dominant or black spruce mixed wood | 61 | | 41 |
| Black spruce lowland | 41 | 101 | 41 * |
| Treed bog and fen | Permanent | | Permanent |

* We assumed that harvested forest would require 40 years to regain suitable habitat status.

For every combination of 10-year forest age class and land cover type associated with suitable habitat (i.e., useable, preferred, or refuge habitat), if it was present in site $n$ in period $t$, a score of 1 times the proportion of that combination was assigned, and a total habitat suitability value for a site was estimated as the sum of these scores. In turn, the total habitat suitability value for a forest patch was estimated as a sum of these scores from the individual sites within a patch and then rescaled to a 0–1 range (Figure 3b). Sites may experience anthropogenic disturbances that are undesirable for caribou populations. We adjusted the habitat amounts $b_{nit}$ in each site $n$ using a habitat intactness coefficient that accounted for all human-mediated disturbances in $n$. We estimated the intactness as the proportion of the site's area outside of a 500-m buffer around any human disturbances, as defined in Environment Canada's scientific assessment for caribou [8]. We then used Equation (17) to estimate the amount of habitat $B_j$ in patches $j$ (Figure 3b,c).

We also factored in the impact of permanent roads on the suitability of caribou habitat. We used estimates of how well caribou population recruitment rates are predicted by the buffer width around linear features [9] to determine the distance to a road where an impact on caribou recruitment is most appreciable. The impact of roads is felt as much as 2 km away [9]. For each patch $j$, we calculated the permanent road impact factor as the proportion of the patch's area outside of a 2-km buffer around permanent roads in $j$ (Figure 3d) and used this factor to adjust the habitat amounts $B_j$ (Figure 3e).

The harvest scheduling model required estimates of delivery costs, volumes of timber, and net revenues for the set of harvest prescriptions $I$. We used the CanVec road database [74] to estimate hauling costs, assuming an on-site harvest cost of Cdn \$15 m$^{-3}$ and delivery of timber to the mill in Dryden from the Trout Lake and Lac Seul FMUs, or to the FP Resolute mill in Thunder Bay from the Caribou FMU (Figure 3f). The hauling costs were based on typical estimates for northern Ontario conditions [75] and included the delivery cost with a hauling rate of \$90-hr$^{-1}$, assuming a 40-m$^3$ truckload, one-hour waiting time, and an overhead cost of \$4 m$^{-3}$. The starting values for stand age, timber volume, land cover composition, anthropogenic disturbances, and the extent of the harvestable area were estimated from Ontario's Forest Resource Inventory database [76] (Figure 3g–i). To estimate the future timber yields in the harvest prescriptions, we used a set of yield curves for northwestern Ontario from [77]. These projected forest area estimates were adjusted by the predicted annual losses due to forest fires using fire regime zones from [78]. The minimum age of harvest was set to 70 years. Harvest planning often includes criteria to maintain sustainable long-term use of forest resources without overharvesting. We assumed that the region-wide mean forest age at the end of the planning horizon should be equal to or greater than 80 years. We set the harvest planning horizon $T$ to 100 years with $10 \times 10$-year time periods.

We first solved the follower's problem for all combinations of $D$ FMUs $\times$ $C$ protection levels ranging from 5% to 90% of the FMU's area. Next, we solved the bi-level leader's problems 2, 4, and 6 using the precomputed follower's optima. We also solved the single-level problems 1, 3, 5, and 7. To explore the trade-offs between problems 1–7, we plotted the solutions in dimensions of total harvest revenue and the amount of protected habitat (Figure 4). Additionally, we examined the spatial patterns of protected sites in the optimal solutions for each of the problems.

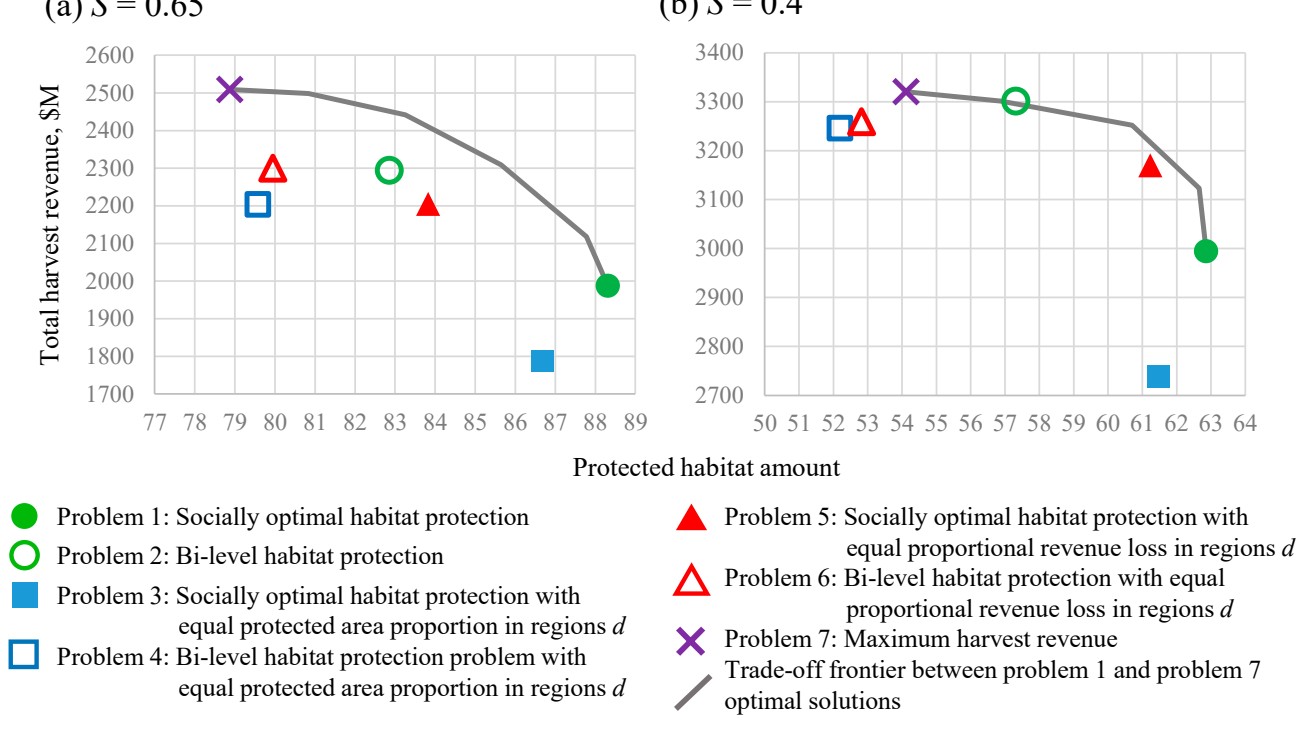

**Figure 4.** Trade-offs between problem 1–7 optimal solutions in dimensions of total harvest revenue and the amount of protected habitat in the range area: (**a**) optimal solutions for the protected habitat target $S = 0.65$; (**b**) optimal solutions for $S = 0.4$. The protected habitat amount is depicted as the dimensionless scores.

## 3. Results

### 3.1. Trade-Off between Maximizing Habitat Protection and Maximizing Harvest Revenue

We compared the optimal solutions for problems 1–7 for two habitat protection targets $S$ equal to 40% and 65% of the range area. Based on previous caribou studies in Canada, the National Recovery Strategy for caribou identified 65% undisturbed habitat in a caribou range as a conservation management threshold that provides a measurable probability (60%) for local caribou populations to be self-sustaining [8,9]. Alternatively, a 40% protection target is close to the area that would be left unharvested in the maximum sustainable harvest scenario under the current harvest assumptions (i.e., when constraints (36) and (37) limit the harvested volume to prevent overharvesting). Figure 4 depicts the optimal solutions for each problem in dimensions of total harvest revenue and protected habitat amount and Figures 5 and 6 show the maps of protected sites for problems 1–7.

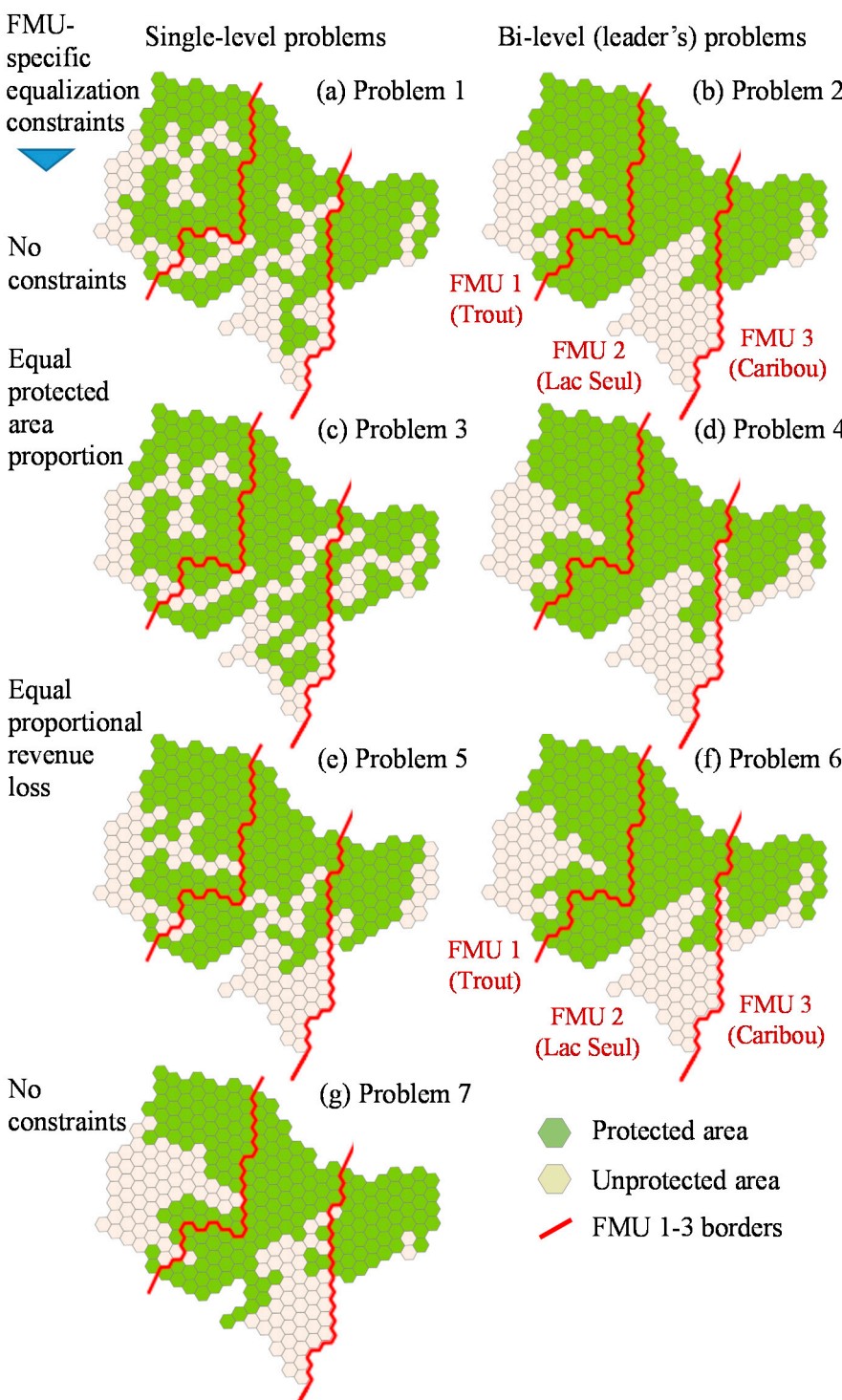

**Figure 5.** Habitat protection patterns in problem 1–7 solutions, *S* = 0.65.

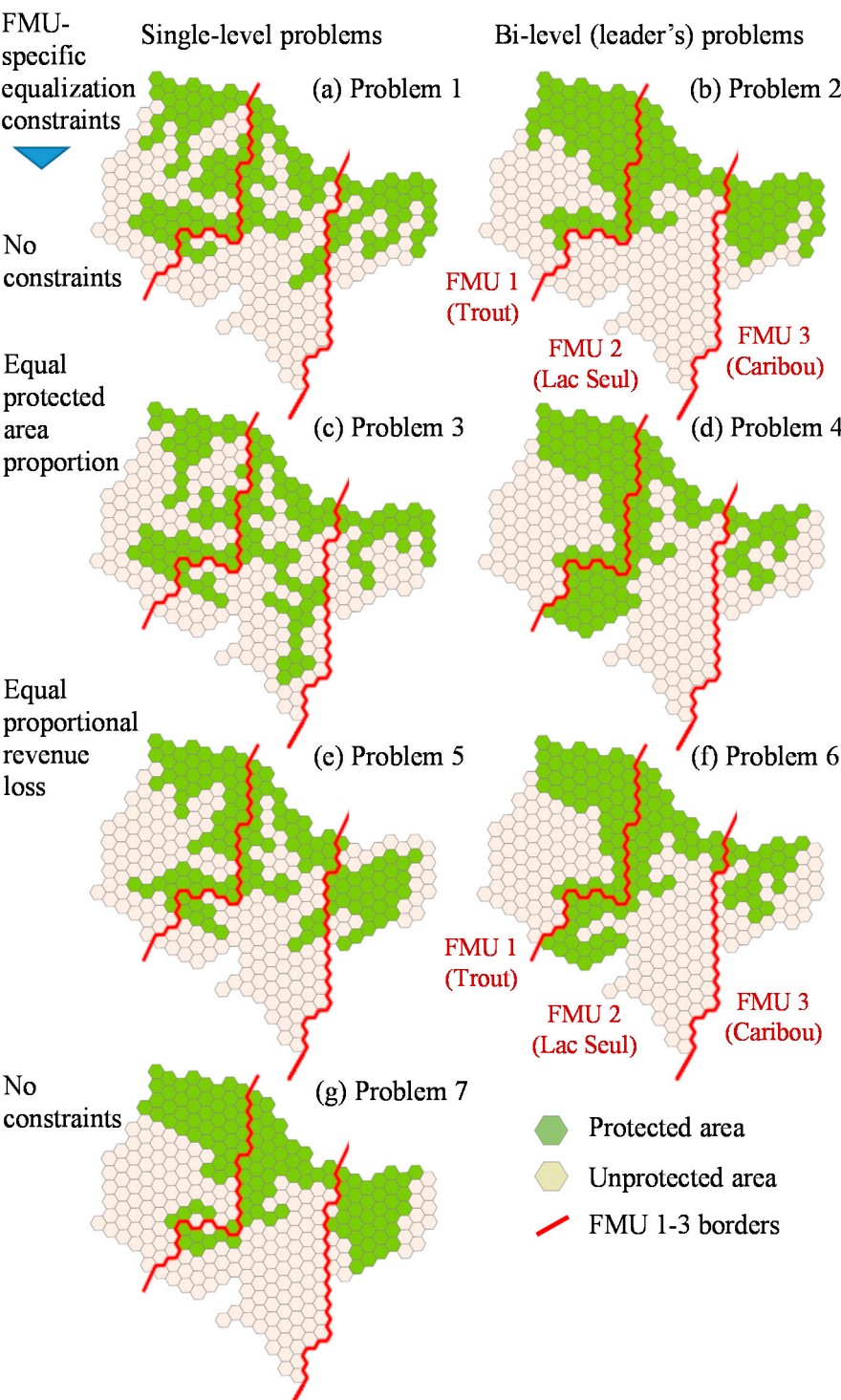

**Figure 6.** Habitat protection patterns in problem 1–7 solutions, *S* = 0.4.

As expected, problem 1 protected the largest amount of habitat, and problem 7 generated the highest revenue from harvest (Figure 4, Table 4). These solutions delineate the endpoints of the trade-off between maximizing harvest revenue and maximizing the amount of protected habitat within the study area. The trade-off between problem 1 and problem 7 solutions is a convex frontier (Figure 4). This represents an upper bound: no solutions have better combinations of harvest revenue and protected habitat amount within the study area than the solutions on the frontier.

**Table 4.** Harvest revenue, protected habitat amount and FMU-specific harvest revenue loss ratios * between the revenue is protection and no-protection scenarios.

| Problem | Protected Habitat Amount | Total Harvest Revenue | Protected Area Proportion Target | Actual Protected Area Proportion | Protected Area Proportion per Region | | | | Harvest Revenue Loss Ratio per Region * | | | |
|---|---|---|---|---|---|---|---|---|---|---|---|---|
| | | | | | FMU 1 Trout | FMU 2 Lac Seul | FMU 3 Caribou | Min-Max Range | FMU 1 Trout | FMU 2 Lac Seul | FMU 3 Caribou | Min-Max Range |
| Protected area proportion target $S = 0.65$ | | | | | | | | | | | | |
| Problem 1: Socially optimal habitat protection | 88.3 | 1987.4 | 0.65 | 0.67 | 0.668 | 0.596 | 0.909 | 0.313 | 0.457 | 0.335 | 0.715 | 0.38 |
| Problem 2: Bi-level habitat protection | 82.9 | 2294 | 0.65 | 0.67 | 0.632 | 0.631 | 0.89 | 0.259 | 0.289 | 0.317 | 0.694 | 0.405 |
| Problem 3: Socially optimal habitat protection with equal protected area % in FMUs $d$ | 86.7 | 1786.9 | 0.65 | 0.67 | 0.674 | 0.675 | 0.673 | 0.002 | 0.469 | 0.463 | 0.477 | 0.014 |
| Problem 4: Bi-level habitat protection problem with equal protected area % in FMUs $d$ | 79.6 | 2203.7 | 0.65 | 0.66 | 0.655 | 0.655 | 0.654 | 0.001 | 0.33 | 0.364 | 0.172 | 0.192 |
| Problem 5: Socially optimal habitat protection with equal proportional revenue loss in FMUs $d$ | 83.8 | 2205.1 | 0.65 | 0.66 | 0.655 | 0.621 | 0.749 | 0.128 | 0.345 | 0.338 | 0.338 | 0.007 |
| Problem 6: Bi-level habitat protection with equal proportional revenue loss in FMUs $d$ | 79.9 | 2300 | 0.65 | 0.65 | 0.644 | 0.631 | 0.729 | 0.098 | 0.308 | 0.317 | 0.317 | 0.009 |
| Problem 7: Maximum harvest revenue | 78.9 | 2509.3 | 0.65 | 0.64 | 0.51 | 0.65 | 0.948 | 0.438 | 0.111 | 0.353 | 0.884 | 0.773 |

**Table 4.** *Cont.*

| Problem | Protected Habitat Amount | Total Harvest Revenue | Protected Area Proportion Target | Actual Protected Area Proportion | Protected Area Proportion per Region | | | | Harvest Revenue Loss Ratio per Region * | | | |
|---|---|---|---|---|---|---|---|---|---|---|---|---|
| | | | | | FMU 1 Trout | FMU 2 Lac Seul | FMU 3 Caribou | Min-Max Range | FMU 1 Trout | FMU 2 Lac Seul | FMU 3 Caribou | Min-Max Range |
| | | | | Protected area proportion target $S = 0.4$ | | | | | | | | |
| Problem 1: Socially optimal habitat protection | 62.9 | 2993.9 | 0.4 | 0.42 | 0.478 | 0.294 | 0.6 | 0.31 | 0.224 | 0.004 | 0.336 | 0.34 |
| Problem 2: Bi-level habitat protection | 57.3 | 3308.6 | 0.4 | 0.42 | 0.494 | 0.248 | 0.691 | 0.44 | 0.101 | 0.005 | 0.227 | 0.23 |
| Problem 3: Socially optimal habitat protection with equal protected area % in FMUs $d$ | 61.5 | 2737.9 | 0.4 | 0.41 | 0.414 | 0.415 | 0.409 | 0.01 | 0.202 | 0.165 | 0.132 | 0.07 |
| Problem 4: Bi-level habitat protection problem with equal protected area % in FMUs $d$ | 52.2 | 3245.6 | 0.4 | 0.40 | 0.397 | 0.397 | 0.404 | 0.01 | 0.012 | 0.051 | 0.001 | 0.05 |
| Problem 5: Socially optimal habitat protection with equal proportional revenue loss in FMUs $d$ | 61.2 | 3170 | 0.4 | 0.41 | 0.429 | 0.341 | 0.586 | 0.25 | 0.086 | 0.005 | 0.27 | 0.27 |
| Problem 6: Bi-level habitat protection with equal proportional revenue loss in FMUs $d$ | 52.8 | 3259 | 0.4 | 0.40 | 0.397 | 0.4 | 0.421 | 0.02 | 0.013 | 0.042 | 0.001 | 0.04 |
| Problem 7: Maximum harvest revenue | 54.1 | 3320.5 | 0.4 | 0.40 | 0.384 | 0.311 | 0.661 | 0.35 | 0.002 | 0.007 | 0.133 | 0.13 |

* Revenue loss ratio denotes the total revenue in the solution to the stated problem with habitat protection divided by the total revenue in a no-protection, harvest-only solution (which was run separately).

All equal protected areas and equal revenue loss solutions (problems 3–6) are located below the frontier. This reflects the associated penalties for imposing either equal protected area proportion or equal revenue loss constraints on the objective value. The solutions for problems 2 and 5 are closest to the frontier, i.e., their performance is close to the theoretical upper bound. The problem 2 solution occupies an intermediate position between the problem 1 and 7 solutions. However, while it is located closer to the frontier than the problem 5 solution, the problem 2 solution does not address the issue of unequal revenue losses among the forestry companies.

The optimal solutions for problems 3 and 4, which enforce equal protected area proportions in each FMU, and the solution for bi-level problem 6, which enforces equal pro-portional harvest revenue loss, are appreciably farther from the frontier than the solutions for problems 2 and 5. With respect to problems 3 and 4, the penalty of protecting equal proportions of the area in each FMU is high and does not guarantee equitability in revenue losses. Problem 6 shares the equal revenue loss constraint with problem 5 but shows worse performance in dimensions of protected habitat amount and harvest revenue.

The most notable difference between the single-level planner's models 1 and 7 and the bi-level models is that the single-level planner's models give exact prescriptions of forest sites to protect or harvest. These regulatory prescriptions are relaxed in the bi-level models. As a result, the optimal solutions are not as efficient as the single-level problem solutions for problems 1 and 2 and the frontier in Figure 4. The bi-level optimal solutions are below the frontier (Figure 4) because the followers (timber companies) alter the harvest allocation to maximize profits, which is not in the best interest of the regulator.

For the same habitat protection target *S*, the bi-level problem solutions (problems 2, 4, and 6) protected lower amounts of habitat but generated higher revenue than the socially optimal problem solutions (problems 1, 3, and 5). The bi-level problem 2 protects less habitat than problem 1 because the regulator gives decision-makers in FMUs the flexibility to decide where and when to harvest forest to maximize revenue, subject only to a constraint on the protected area proportion and maintaining area-wide forest age and even harvest flow over time. There is no constraint on the locations or specific amounts of habitat that needs protection. The bi-level problems 4 and 6 have additional constraints that further reduce the amount of protected habitat with little effect on harvest revenue. Note that the equal protected area and equal revenue loss solutions for problems 2 and 4–6 are located closer to the frontier in the scenarios with the smaller protection target (40%) than in the scenarios with the larger protection target (65%). This is because the smaller protected area offers more flexibility in allocating harvest, and therefore the revenue values for these solutions are closer to the theoretical upper bound.

### 3.2. General Habitat Protection Patterns

In the optimal solutions for problem 1 at both protection targets, the protected area included patches with large amounts of intact habitat and avoided patches with permanent roads (Figures 5a and 6a). In the bi-level problem 2 solutions, the protected area was more spatially compact and included some patches with low-quality habitat. Accordingly, the unprotected area included sites with low harvest costs due to their proximity to access roads in the western and southern portions of the range (Figure 5b).

When the habitat protection target was low (*S* = 0.4), the bi-level models 2, 4, and 6 concentrated the protected sites along the range's northern border and in its center, where there are large amounts of high-quality habitat (Figure 6b,d,f). The patterns of protected habitat in the socially optimal models 1, 3, and 5 were more spatially complex than those in the bi-level problem solutions because they tended to avoid permanent roads while still including pockets of high-quality habitat in the central part of the range (Figure 6a,c,e). On a per-hectare basis, the bi-level problem solutions allocated lower-quality habitat to the protected area than the socially optimal problem solutions (Table 4).

Typically, the bi-level models excluded easily accessible sites with large volumes of productive forest (but also large amounts of high-quality habitat) from protection. In

general, the solutions for problems 1 and 2 allocated a protected area proportion that was reasonably close to the *S* target in FMU 1, smaller than the target in FMU 2, and larger than the target in FMU 3 (Table 4).

Adding the equal protected area constraint in problems 3 and 4 moderately changed the patterns of protected patches, yet their general configurations of protected patches resembled the configurations in problems 1 and 2 (Figure 5c,d and Figure 6c,d). Similar changes were observed for the problem 5 and 6 solutions that enforced equal proportional revenue loss among the FMUs. The general strategy was to protect areas of prime habitat in the central part of the range and near the northern borders while increasing harvest intensity in the southern and western parts, where there are high densities of permanent roads.

We also examined the dependencies between the protected area proportion $S_d$ and harvest revenue for each FMU (Figure 7). Revenue increases when the protected area proportion decreases and stabilizes when the protected area proportion target drops below 0.3 (i.e., where the curves in Figure 7 become vertical). From this point, a further decrease in the protected area does not lead to an increase in revenue. This is a result of the other constraints in the harvest model, such as even harvest flow, minimum harvest age, and minimum average forest age, which limit the total area that can be harvested. Figure 7 shows solution curves for two distinct scenarios that maximize either the amount of protected habitat (dotted lines) or harvest revenue (solid lines) in an FMU. These solutions are based on problems 1 and 7 solved for individual FMUs *d*.

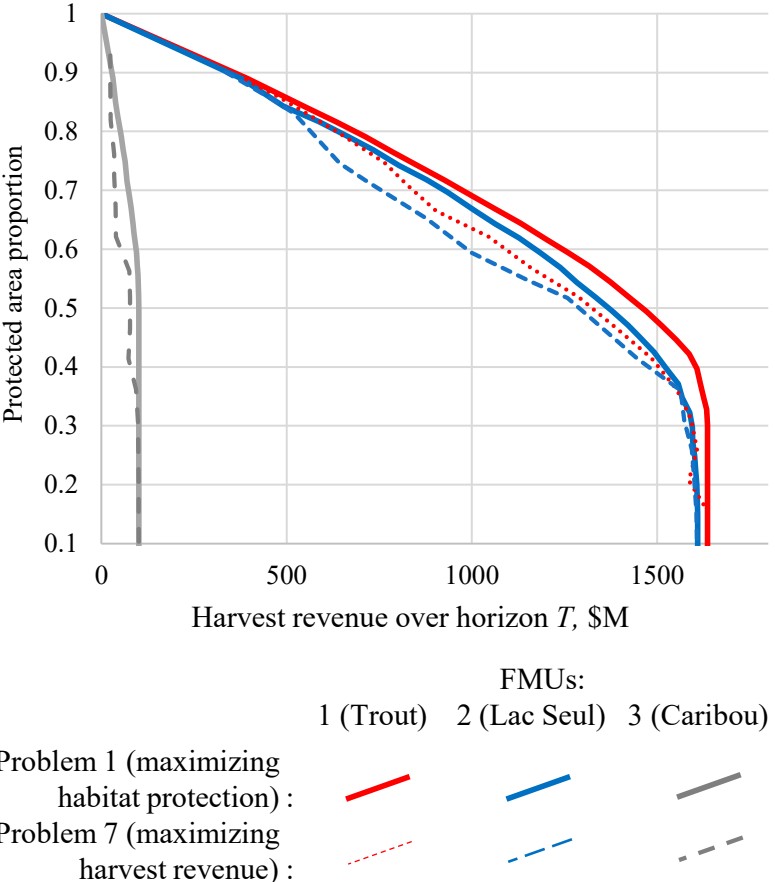

**Figure 7.** The proportion of protected habitat area exempt from harvest in the FMU area vs. total harvest revenue for FMUs 1–3. The curves show the solutions for problems 1 and 7, which represent the endpoints of the trade-off frontier (see Figure 4) between solutions that maximize the amount of protected habitat (Problem 1; solid lines) and solutions that maximize harvest revenue (Problem 7; dotted/dashed lines).

On a proportional basis, FMU 3 (Caribou FMU) experiences the lowest impact of habitat protection on harvest revenue and FMU 2 (Lac Seul FMU) has the highest impact (Table 4). For example, in the solutions for problems 5 and 6, which both included an equal revenue loss constraint, a larger area proportion is protected in FMU 3 than in FMUs 1 and FMU 2. This is because the proportional revenue loss increases with the area of low-cost productive forest set aside for protection. FMU 2 (Lac Seul FMU) has the largest area and FMU 3 has the smallest area of low-cost productive forest, so it is optimal to harvest more timber in FMU 2 than in FMUs 1 and 3.

## 4. Discussion

Industrial timber harvesting is one of the most notable human activities in Canadian boreal forests and has significant economic and social potential. However, with mounting pressures to protect caribou populations in areas of active forestry and more stringent habitat protection regulations [3,31,32], it is imperative to understand how regulation, particularly the whole-range protection of caribou populations, may affect the forestry players operating in an area of planned protection. In our study, we considered a wildlife conservation planning problem in which a central regulator maximizes habitat protection at the caribou range level while industry decision-makers in forest management units within the range aim to maximize harvest revenue. Our approach makes it possible to integrate the aspirations of multiple economic players, acting simultaneously with little or no coordination, into regulatory decisions regarding conservation efforts.

Our work helps address two essential questions related to the development of whole-range caribou protection measures. First, how impactful is the whole-range protection on individual forestry players, and what are their likely responses to regulatory policies? Second, what is the effective regulatory setting to ensure a fair playing field for multiple economic agents operating in the area of planned protection? Related to this, we investigate the impact of equitability measures for individual economic agents on habitat protection and harvest benefits. In our case study, we found that accounting for the anticipated responses of forestry players to habitat protection regulation has a moderate impact on the habitat protection efficacy and harvest benefits at the range level (as shown in our bi-level problem 2 solutions). However, attaining equitable impacts of habitat protection on individual forestry players is likely to increase these range-level impacts. Without equalization criteria, the solutions for both problems 1 and 2 allocated unequal proportions of protected area in each FMU (Table 4). By comparison, the three FMUs had almost equal proportions set aside for protection in the problem 3 and 4 solutions with equalization constraints. However, equalization comes at a price: The problem 3 solutions had the lowest harvest revenue among all problem solutions, and problem 4 had the lowest amount of protected habitat in the solutions with $S = 0.4$ (Figure 4b) and the second-lowest amount in the solutions with $S = 0.65$ (Figure 4a). The problem 5 and 6 solutions with equal proportional revenue loss demonstrated similar behavior.

The logistical challenge of reducing forest disturbance to levels prescribed by federal and provincial endangered species regulations has been acknowledged for boreal forest areas throughout Canada [27,79,80]. For example, in Ontario, forestry operations were exempted from the province's Endangered Species Act and are not yet compliant with the Federal Species-at-Risk Act [27]. Our proposed approach facilitates the evaluation of habitat protection options in areas of active forestry and could help achieve wider compliance with provincial and federal caribou recovery strategies [7,8]. Incorporating the market-oriented objectives of FMU decision-makers into an agency planner's decisions may establish financial or reputational incentives that enable sufficient protection of critical wildlife habitat [79]. Furthermore, forest companies may be motivated to support caribou protection when seeking Forest Stewardship Council (FSC) certification [81–83] for their management areas (which requires a forest management plan that ensures the persistence of caribou populations). The primary purpose of FSC certification is to provide a level

of assurance to consumers that forest products have been produced from healthy forests while respecting natural forest processes and biodiversity.

In order to build whole-range habitat protection into an economically and environmentally robust practice, government and industry need to work together to remove pervasive perceptions among forest industry players that habitat protection regulation is exceedingly detrimental to their interests. Our modeling approach helps provide a starting point to find common ground between the forestry sector and government regulators on the adoption of whole-range caribou protection measures in Canada. Such collaboration between governments and the forest sector is crucial to properly understand the application of whole-range protection and to capture the full suite of benefits it can offer.

Our study illustrates some advantages of a bi-level, game-theoretic approach. Our socially optimal problem 1 implies that the regulator is able and willing to allocate protected areas down to the patch level to maximize its habitat protection utility. The problem's optimal solution achieves the maximum habitat protection utility, but as a practical matter, a problem 1 formulation is unlikely to be adopted because it ignores the interests of forestry companies operating in the area of concern. An opposite approach is depicted in problem 7. The objective maximizes the sum of the revenues for all companies subject to the protection area target set by the regulator. This scenario provides an upper bound for the best achievable sum of the companies' profits but is unlikely to be adopted by the regulator due to its weak capacity to protect high-quality caribou habitat. Moreover, the solution that maximizes aggregate revenue may not be perceived as fair if the burden of achieving the target level of habitat protection falls within one FMU. The bi-level scenarios, as depicted by problems 2, 4, and 6 may be a better option because they assume that the regulator knows the utility preferences of the companies and endogenizes their decisions concerning their preferences. Note that socially optimal problem solutions showed lower harvest revenue than the bi-level problem solutions (Table 4). This is because the bi-level solutions anticipate the profit-maximizing strategy of the forestry companies within their FMUs and produce a more cost-effective harvest plan. The bi-level problem also sets fewer constraints on harvesting within FMUs. It allows timber companies to choose the forest patches to protect to meet the assigned target while maximizing harvest revenue.

Our current model can potentially be extended in several ways. First, our analysis focuses on a single-species regulatory policy (i.e., woodland caribou). This is a reasonable assumption given the threatened status of caribou populations in Canada. However, there are also situations where wildlife protection policies need to include other threatened species (which may prefer other habitat types than caribou). In such a case, the model can be extended to incorporate habitat data for multiple species and maximize the protection of habitat for all of these species. Potentially, the model could include other environmental guidelines that regulate harvesting practices in the region [84] and/or impose harvest limits set by historical harvest levels or an annual allowable cut level. Incorporating these aspects could be the focus of future work.

**Author Contributions:** Conceptualization: D.Y., R.G.H.; optimization model development, D.Y., R.G.H.; formal analysis, D.Y., R.G.H., R.S.R., F.H.K., A.R., N.L.; data provision: D.Y., R.S.R., A.R.; data curation and visualization, N.L.; writing—original draft preparation, D.Y.; all authors contributed to manuscript review and editing; supervision and funding acquisition, D.Y. All authors have read and agreed to the published version of the manuscript.

**Funding:** The funding for this work was provided by Natural Resource Canada's Cumulative Effects Program.

**Acknowledgments:** This work was supported by Natural Resource Canada's Cumulative Effects Program and in part by the U.S. Department of Agriculture, Forest Service.

**Conflicts of Interest:** The authors declare no conflict of interest.

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
