# Peer review of "Balancing Large-Scale Wildlife Protection and Forest Management Goals with a Game-Theoretic Approach"

_forests, doi:10.3390/f12060809_

Round 1

Reviewer 1 Report

This manuscript presents an approach to policy making that simultaneously improves wood production and ecosystem conservation through a two-step optimization. Although the method is explained in sufficient detail, the overall outline is difficult to grasp and hinders the reader's understanding. For this reason, I suggest that the following modifications.

  1. Move Table 2 and its description to the beginning of Materials and Methods to make it clear what the optimization will be for in this paper.
  2. L167: The author states that connectivity is necessary, but the reference is abrupt and obscure why that is. An overview of the protected and unprotected areas in the study area and their requirements should be added.
  3. I don't understand what node 0 indicates. It is necessary to show the position of node 0 in the research area, including specific examples.
  4. I don't understand the legend of Figure 7.
  5. Only Problem 1 and Problem 7 are mentioned in the discussion. The results of the study need to be discussed more comprehensively.
  6. A discussion of the potential use of the methods in this paper in policy making is required. The reader would benefit greatly from a discussion of what issues might arise in practical application, and what developmental research might be envisioned.

I hope these comments will be helpful.

Author Response

Reply to Reviewers’ Comments:

Reviewer 1:

Move Table 2 and its description to the beginning of Materials and Methods to make it clear what the optimization will be for in this paper.

As suggested, we have moved Table 2 closer to the beginning of the Material and Methods section.

L167: The author states that connectivity is necessary, but the reference is abrupt and obscure why that is. An overview of the protected and unprotected areas in the study area and their requirements should be added.

Controlling connectivity between the unprotected patches prevents the creation of isolated pockets of unprotected forest surrounded by protected area. Maintaining connectivity between undisturbed protected patches allows caribou to move and access their strongly preferred habitat at lower risk of encountering predators. We have added clarifying text with corresponding citations.

I don't understand what node 0 indicates. It is necessary to show the position of node 0 in the research area, including specific examples.

Node 0 is an auxiliary node and does not have a geographical location. As depicted in Fig. 2a, node 0 is connected to all nodes j and is used to inject the flow into the network of protected habitats via arcs 0j to ensure its connectivity. The connectivity between the protected (or unprotected) nodes j is achieved by injecting the flow into one protected node and ensuring that all other protected nodes receive flow from that node (or likewise for unprotected nodes). Injection of the flow to a single node from node 0 ensures that all nodes that receive flow from that node form a single connected graph. The text has been edited to make this clearer.

I don't understand the legend of Figure 7.

We have modified the legend and description of Figure 7. In short, the curves depict optimal solutions to problems 1 and 7 in dimensions of the protected habitat area proportion and total harvest revenue in the FMU area. We have only shown problems 1 and 7 as depicting the end points of the trade-off frontier (see Fig. 4) between solutions that maximize the amount of protected habitat (problem 1; solid lines) and solutions that maximize harvest revenue (problem 7; dotted/dashed lines).

Only Problem 1 and Problem 7 are mentioned in the discussion.

We have mentioned other problems in our significantly revised discussion section, but also want to make it clear that our intention was to keep the discussion focused on general and potential practical application issues only, which explains any emphasis on problems 1 and 7 as the most illustrative among the set of seven problems. To limit the amount of text and avoid repetitions, we provided brief descriptions of key findings and observations with respect to all of problems in the results section, where we could put them in an appropriate context.

The results of the study need to be discussed more comprehensively. A discussion of the potential use of the methods in this paper in policy making is required. The reader would benefit greatly from a discussion of what issues might arise in practical application, and what developmental research might be envisioned.

We have rewritten the discussion section and added text elaborating on the potential utility of the proposed approach. Given that the study presented seven optimization problems, we chose to group brief descriptions and interpretations of the key findings from all of the problems in the results section, so that we could keep the discussion section more general and focused on the broader policy implications. Given the large number of presented models, we felt this approach was reasonable.

Reviewer 2 Report

This interesting manuscript offers a valuable decision-support tool to aid the management of the trade-off between resource extraction and wildlife conservation. The study improves understanding of how timber harvesters can maximize their profits whilst remaining within the conservation regulations set by the government, which offers a means of reducing the conflict between wildlife conservation and timber harvesting for the benefit of both stakeholder groups. I can see there being considerable attention in using this methodology to address competing demands between stakeholders, in particular for conflicts between wildlife conservation and resource extraction.

Your manuscript is well written and logically set out. You will see below that I had some thoughts on how certain parts of the manuscript could be clarified, which I hope that you will find helpful.

Detailed comments:

Line 36: To aid readers who may be less familiar with this system, I suggest adding "(>50°N Latitude)" after the phase "In boreal Canada".

Lines 36-37: The negative association between active forestry and Caribou distribution was nicely demonstrated in the following paper, which could be cited here: Schaefer, J.A. (2003). Long‐term range recession and the persistence of caribou in the taiga. Conservation Biology17 (5), 1435-1439.

Lines 45-46: "Indeed, caribou constantly move over long distances as part of their foraging and seasonal behavior." This sentence should be accompanied by an appropriate supporting reference, e.g. Ferguson, S.H. & Elkie, P.C. (2004). Seasonal movement patterns of woodland caribou (Rangifer tarandus caribou). Journal of Zoology262 (2), 125-134.

Lines 51-55: I think that the problem could be spelled out more explicitly here. Crucially, the issue is that there are competing pressures on land use between those people who want to prioritize caribou conservation, and those people who want to prioritize timber production. This gives rise to a 'conservation conflict' (sensu Redpath et al. 2015) between these groups. Identifying the problem in these terms will help to link your current study in with the wider literature on this topic. I recommend inserting this information on line 52 between the first sentence that mentions the trade-off between range protection and forest harvesting, and before the sentence which mentions the active research on this topic, as the new information bridges these two points. Redpath, S.M., et al. (2015). Conflicts in conservation: navigating towards solutions. Cambridge University Press.

Lines 98-99. "In Canada, provincial government authorities issue these licenses". Does the provincial government also have statutory responsibility for caribou conservation, or is that within the remit of the federal government? It would be useful to clarify this here.

Line 154: Can a scale bar be added to Figure 1?

Lines 288-289: The scientific names of these competitor species (deer and moose) and the two predator species (black bears and wolves) should be given here.

Line 297: "Over horizon T, no harvesting is allowed in patches j selected for protection". Presumably, illegal logging within protected areas is not considered to be an issue in this system? You could refer to this earlier analysis, which supports that point: Li, R., et al. (2008). Long-term effects of eliminating illegal logging on the world forest industries, trade, and inventory. Forest Policy and Economics10 (7-8), 480-490.

Lines 465-466: Please state which version of the General Algebraic Modeling System was used in your analysis. This should be done in citation details on lines 856-857.

Lines 480-482: "The area was moderately fragmented by logging and road construction, with timber delivered to pulp and paper mills in the nearby cities of Thunder Bay and Dryden, ON." Are there any published articles or reports that could be cited to support this point about fragmentation?

Line 497: In the legend for Table 3, the units for age should be specified (presumably 'years').

Line 543: Typo - there are two periods at the end of this sentence, please delete one.

Lines 563-566: What are the units for the protected habitat amount on the x-axis in Figure 4? Please state this either on the figure itself or in the legend.

Line 656: Should this be Figure 6? Figure 5 was already presented (see the legend line 642). Please clarify and reorder the figures as required.

Line 697: The acronym "SARA" should be defined on first use here.

Lines 856-857: "GAMS (GAMS Development Corporation). General Algebraic Modeling System (GAMS) Washington, DC, USA, 2019. http://www.gams.com (accessed 1 Feb 2018)." Please state the version that was used in your study.

Author Response

Reply to Reviewers’ Comments:

Reviewer 2:

Detailed comments:

Line 36: To aid readers who may be less familiar with this system, I suggest adding "(>50°N Latitude)" after the phase "In boreal Canada".

The sentence has been edited as the reviewer suggested.

Lines 36-37: The negative association between active forestry and Caribou distribution was nicely demonstrated in the following paper, which could be cited here: Schaefer, J.A. (2003). Long‐term range recession and the persistence of caribou in the taiga. Conservation Biology, 17 (5), 1435-1439.

We have added a citation to Schaefer 2003 following the reviewer’s suggestion

Lines 45-46: "Indeed, caribou constantly move over long distances as part of their foraging and seasonal behavior." This sentence should be accompanied by an appropriate supporting reference, e.g. Ferguson, S.H. & Elkie, P.C. (2004). Seasonal movement patterns of woodland caribou (Rangifer tarandus caribou). Journal of Zoology, 262 (2), 125-134.

We have added the citations suggested by the reviewer.

Lines 51-55: I think that the problem could be spelled out more explicitly here. Crucially, the issue is that there are competing pressures on land use between those people who want to prioritize caribou conservation, and those people who want to prioritize timber production. This gives rise to a 'conservation conflict' (sensu Redpath et al. 2015) between these groups. Identifying the problem in these terms will help to link your current study in with the wider literature on this topic. I recommend inserting this information on line 52 between the first sentence that mentions the trade-off between range protection and forest harvesting, and before the sentence which mentions the active research on this topic, as the new information bridges these two points. Redpath, S.M., et al. (2015). Conflicts in conservation: navigating towards solutions. Cambridge University Press.

We have added text in lines 51-55 that acknowledges this issue as an example of a “conservation conflict” when land use policies are subject to the pressures of competing objectives, such as prioritizing wildlife conservation and prioritizing timber production. We also want to note a particular feature of the issue in Canada, namely that timber harvesting occurs on Crown lands and so both habitat protection and harvest are regulated by the provincial ministry of natural resources. Thus, in our case we are presented with a problem of developing a regulatory conservation policy that accounts for economic behaviour of multiple agents operating within a shared area where protection is planned.

Lines 98-99. "In Canada, provincial government authorities issue these licenses". Does the provincial government also have statutory responsibility for caribou conservation, or is that within the remit of the federal government? It would be useful to clarify this here.

Yes, the provincial government has statutory responsibility for caribou conservation as an endangered species, according to the provincial Endangered Species Act. We have edited the text accordingly.

Line 154: Can a scale bar be added to Figure 1?

We have added a scale bar to Figure 1.

Lines 288-289: The scientific names of these competitor species (deer and moose) and the two predator species (black bears and wolves) should be given here.

We have added the scientific names of deer, moose and predator species (black bears and wolves) to the text.

Line 297: "Over horizon T, no harvesting is allowed in patches j selected for protection". Presumably, illegal logging within protected areas is not considered to be an issue in this system? You could refer to this earlier analysis, which supports that point: Li, R., et al. (2008). Long-term effects of eliminating illegal logging on the world forest industries, trade, and inventory. Forest Policy and Economics, 10 (7-8), 480-490.

Illegal logging within protected areas is not an issue in Canadian boreal forests due to sparse population, difficult road access, lack of illegal timber markets and a well-functioning law enforcement system, and so was not considered in the study. There are no records of illegal logging in national/provincial protected areas in boreal Ontario.

Lines 465-466: Please state which version of the General Algebraic Modeling System was used in your analysis. This should be done in citation details on lines 856-857.

We have added the GAMS version number (33) to the text.

Lines 480-482: "The area was moderately fragmented by logging and road construction, with timber delivered to pulp and paper mills in the nearby cities of Thunder Bay and Dryden, ON." Are there any published articles or reports that could be cited to support this point about fragmentation?

This was meant as a general statement, and because our mention of fragmentation isn’t especially germane here, we have removed it.

Line 497: In the legend for Table 3, the units for age should be specified (presumably 'years').

We have edited the Table 3 title to specify the units for minimum age (years).

Line 543: Typo - there are two periods at the end of this sentence, please delete one.

Deleted.

Lines 563-566: What are the units for the protected habitat amount on the x-axis in Figure 4? Please state this either on the figure itself or in the legend.

The habitat amount was depicted as dimensionless score values.

Line 656: Should this be Figure 6? Figure 5 was already presented (see the legend line 642). Please clarify and reorder the figures as required.

Yes, this should be Fig.6 – we renumbered the figure caption. 

Line 697: The acronym "SARA" should be defined on first use here.

SARA means Federal Species-at-Risk Act. The acronym has been spelled out.

Lines 856-857: "GAMS (GAMS Development Corporation). General Algebraic Modeling System (GAMS) Washington, DC, USA, 2019. http://www.gams.com (accessed 1 Feb 2018)." Please state the version that was used in your study.

We have added the GAMS version number (v.33).